# PROTAP: A BENCHMARK FOR PROTEIN MODELING ON REALISTIC DOWNSTREAM APPLICATIONS

## ABSTRACT

Recently, extensive deep learning architectures and pretraining strategies have been explored to support downstream protein applications. Additionally, domain-specific models incorporating biological knowledge have been developed to enhance performance in specialized tasks. In this work, we introduce **Protap**, a comprehensive benchmark that systematically compares backbone architectures, pretraining strategies, and domain-specific models across diverse and realistic downstream protein applications. Specifically, Protap covers five applications: three general tasks and two novel specialized tasks, i.e., enzyme-catalyzed protein cleavage site prediction and targeted protein degradation, which are industrially relevant yet missing from existing benchmarks. For each application, Protap compares various domain-specific models and general architectures under multiple pretraining settings. Our empirical studies imply that: (i) Though large-scale pretraining encoders achieve great results, they often underperform supervised encoders trained on small downstream training sets. (ii) Incorporating structural information during downstream fine-tuning can match or even outperform protein language models pretrained on large-scale sequence corpora. (iii) Domain-specific biological priors can enhance performance on specialized downstream tasks. Code is publicly available at `https://anonymous.4open.science/r/protap-1CC5`.

## 1 INTRODUCTION

Proteins serve as the central executors of biological activities, regulating a wide range of critical biological processes through their complex three-dimensional structures and dynamic properties. A precise understanding of protein function and interactions is critical across many applications (Wong et al., 2024; Anfinsen, 1973). For instance, directed evolution of enzymes can endow proteins with novel functions (Austin et al., 2018). Accurate prediction of protein–ligand interactions (PLIs) can largely accelerate drug discovery (Sadybekov et al., 2022; Zhang et al., 2025; Wu et al., 2024). These application areas underscore the immense potential of deep learning in protein analysis.

Various deep learning approaches leveraging protein sequences and structures have been developed for protein-related applications. For instance, general sequence models such as LSTM (Hochreiter & Schmidhuber, 1997) and Transformer (Vaswani et al., 2017) have been employed to extract amino acid sequence patterns for tasks like stability landscape prediction (Rao et al., 2019). Geometric Graph Neural Networks (GNNs), including GVP-GNN (Jing et al., 2021) and EGNN (Satorras et al., 2021), have demonstrated remarkable effectiveness in modeling 3D molecular structures. Motivated by these successes, geometric GNNs have been further applied to structure-based protein modeling tasks (Dauparas et al., 2022; Mikhael et al., 2024). Recently, Graphformer (Ying et al., 2021) and Transformer-M (Luo et al., 2023a) have incorporated structural biases into transformer-based architectures. These emerging sequence-structure hybrid models show promising ability in the general protein representation learning.

Recently, the remarkable success of large-scale pretraining models in image and text has inspired similar advances in protein modeling. Masked language modeling has been extended to predict the masked amino acids in protein sequences, resulting in protein language models like ESM series models and ProteinBERT (Rives et al., 2021; Hayes et al., 2025; Brandes et al., 2022). GearNet (Zhang et al., 2023) explores the multi-view contrastive learning on protein structures. Additionally, some

methods, such as OntoProtein (Zhang et al., 2022), leverage functional annotations as supervision signals for pretraining.

Apart from these general model architectures and pretraining tasks for protein modeling, significant progress has been made in developing domain-specific models tailored for various realistic downstream protein applications. For instance, protein function prediction models such as DP-Func (Wang et al., 2025) leverage sequence, structural, and domain-level information to enhance the accuracy of predicting the Gene Ontology annotations of proteins. Additionally, to improve enzyme modeling, UniZyme (Li et al., 2025) integrates the energy frustration matrix and enzyme active-site knowledge into a transformer-based framework. In the domain of proteolysis-targeting chimera (PROTAC) modeling, dedicated methods such as DeepProtacs (Li et al., 2022) and ETProtacs (Cai et al., 2025) have been developed to accurately capture the ternary interactions essential for targeted protein degradation.

Given the extensive variety of general model architectures, pretraining strategies, and domain-specific models, there remains a gap in systematically benchmarking general pretrained models alongside domain-specific models. With numerous architectures and pretraining strategies tailored for real-world downstream tasks, a natural question arises: do existing architectures and strategies exhibit distinct advantages across specific protein applications? However, current benchmarks predominantly focus on specific pretrained model categories, such as protein language models in (Xu et al., 2022; Capel et al., 2022) and geometric GNNs in (Jamasb et al., 2024). As shown in Tab. 1, they lack a comprehensive evaluation of general protein model architectures, pretraining strategies, and domain-specific models across realistic biological applications. To address these gaps, we introduce Protap, a standardized benchmark that systematically compares architectures, pre-training strategies, and domain models on diverse realistic downstream applications. Our contributions are:

- We identify and integrate realistic applications from existing literature and databases to support comprehensive evaluations. In addition to three general applications, Protap introduces two novel specialized applications: enzyme-catalyzed protein cleavage site prediction and targeted protein degradation by PROTACs, which are biological processes not covered by prior benchmarks.
- The proteins covered in the benchmark evaluation are comprehensive and diverse, which cover enzymes, receptors, drugs, etc. The tasks are diverse, which include single protein modeling (function, mutation), interaction modeling such as PLI, enzyme-substrate modeling, and complex interaction process modeling (PROTACs).
- We compare a large number of protein pretraining models and domain models on five protein applications. This offers insights into the development of protein foundation models and the design of domain-specific models for downstream applications.

Table 1: Comparison of benchmark coverage for protein modeling. ♣ indicates the presence of this dimension, while ○ denotes its absence. We compare across three dimensions: applications (specialized vs. general), pretraining tasks, and model architectures (domain-specific vs. general). Comparisons with ProteinGym and ProteinBench are not included because their focuses differ from Protap. ProteinGym (Notin et al., 2023) primarily targets mutation effect prediction with task-specific models. ProteinBench (YE et al., 2025) emphasizes generative design tasks, e.g., backbone design and sequence-structure co-design, which are currently beyond Protap's scope. **We provide a comprehensive literature review (Related Works) in Appendix B.**

| | | Protap (Ours) | PEER (Xu et al., 2022) | ProteinWorkshop (Jamasb et al., 2024) | TAPE (Rao et al., 2019) | ProteinGLUE (Capel et al., 2022) |
|---|---|---|---|---|---|---|
| Specialized Applications | Enzyme-catalyzed protein cleavage site prediction | ♣ | ○ | ○ | ○ | ○ |
| | Targeted protein degradation by Proteolysis-targeting chimeras | ♣ | ○ | ○ | ○ | ○ |
| General Applications | Protein–ligand interactions | ♣ | ♣ | ○ | ○ | ♣ |
| | Function prediction | ♣ | ♣ | ♣ | ○ | ○ |
| | Mutation effect prediction | ♣ | ♣ | ○ | ♣ | ○ |
| Pretraining Tasks | Masked language modeling | ♣ | ○ | ♣ | ♣ | ♣ |
| | Multi-view contrastive learning | ♣ | ○ | ○ | ○ | ○ |
| | Protein family prediction | ♣ | ○ | ○ | ○ | ○ |
| General Architectures | Protein language models | ♣ | ♣ | ○ | ♣ | ♣ |
| | Geometric GNNs | ♣ | ○ | ♣ | ○ | ○ |
| | Sequence-structure hybrid models | ♣ | ○ | ○ | ○ | ○ |
| Domain-specific Architectures | Protein function models | ♣ | ○ | ○ | ○ | ○ |
| | Enzyme domain models | ♣ | ○ | ○ | ○ | ○ |
| | Ternary complexes models | ♣ | ○ | ○ | ○ | ○ |

Table 2: Overview of models and datasets used in Protap.

(a) Pretraining models and domain models in our Protap. Models highlighted in (*) are trained from scratch, while those in (*) use only publicly available pretrained weights.

| Model | Input Modalities | Pretrain Data | #Parameter | Objective | Source |
|---|---|---|---|---|---|
| | | Pretrain Models | | | |
| EGNN | AA Seq & 3D Coord | Swiss-Prot 540k | 10M | MLM, MVCL, PFP | *ICML, 2021* |
| SE(3) Transformer | AA Seq & 3D Coord | Swiss-Prot 540k | 4M | MLM, MVCL, PFP | *NeurIPS, 2020* |
| GVP | AA Seq & 3D Coord | Swiss-Prot 540k | 2M | MLM, MVCL, PFP | *ICLR, 2021* |
| ProteinBERT | AA Seq | Swiss-Prot 540k | 72M | MLM, MVCL, PFP | *Bioinformatics, 2022* |
| D-Transformer | AA Seq & 3D Coord | Swiss-Prot 540k | 3.5M | MLM, MVCL, PFP | *NeurIPS, 2025, ICLR, 2023* |
| ESM-2 | AA Seq | UR50 70M | 650M | MLM | *Science, 2023* |
| ESM Cambrian | AA Seq | UR70, MGnify, JGI 3B | 600M | MLM | *Science, 2025* |
| SaProt | AA Seq & 3D Coord | UR50 40M | 650M | MLM | *ICLR, 2024* |
| | | Domain Specific Models | | | |
| ClipZyme | AA Seq & 3D Coord & SMILES | – | 14.8M | PCS | *ICML, 2024* |
| UniZyme | AA Seq & 3D Coord | Swiss-Prot 11k | 15.5M | PCS | *NeurIPS, 2025* |
| DeepProtacs | AA Seq & 3D Coord & SMILES | – | 0.1M | PROTACs | *Nature Communications, 2022* |
| ETProtacs | AA Seq & 3D Coord & SMILES | – | 5.4M | PROTACs | *Briefings in Bioinformatic, 2025* |
| KDBNet | AA Seq & 3D Coord & SMILES | – | 3.4M | PLI | *Nature Machine Intelligence, 2023* |
| MONN | AA Seq & 3D Coord | – | 1.7M | PLI | *Cell Systems, 2024* |
| DeepFRI | AA Seq & 3D Coord | Pfam 10M | 1.8M | PFA | *Nature Communications, 2021* |
| DPFunc | AA Seq & 3D Coord & Protein Domain | – | 110M | PFA | *Nature Communications, 2025* |

(b) Summary of datasets and metrics for various prediction tasks.

| Application | Category | Data Source | #Train | #Test | Metric |
|---|---|---|---|---|---|
| Pretraining | — | (Consortium, 2019) | 542,378 | — | — |
| Protein Cleavage Site Prediction | Specialized | (Rawlings et al., 2014) | 375 | 92 | AUC, AUPR |
| Targeted Protein Degradation | Specialized | (Ge et al., 2025) | 843 | 209 | Acc, AUC |
| Protein–Ligand Interactions | General | (Luo et al., 2023b) | 11,520 | 2,880 | MSE, Pearson |
| Protein Function Annotation Prediction | General | (Gligorijević et al., 2021; Jamasb et al., 2024) | 23,760 | 2,023 | Fmax, AUPR |
| Mutation Effect Prediction | General | (Notin et al., 2023) | — | 2.4M | AUC, Pearson |

## 2 PROTEIN MODELING IN PROTAP

In this section, we present an overview of how proteins are represented, modeled, and utilized within the Protap benchmark. We begin by outlining the fundamental definitions of protein sequences and structures. We then introduce the three pretraining tasks employed in Protap: masked language modeling, multi-view contrastive learning, and protein family prediction, followed by a summary of the corresponding models. Finally, we describe the downstream applications supported by Protap, which include two specialized tasks, namely Enzyme-Catalyzed Protein Cleavage Site Prediction and Targeted Protein Degradation by Proteolysis-Targeting Chimeras, as well as three general tasks: Protein–Ligand Interactions, Protein Function Annotation Prediction, and Mutation Effect Prediction for Protein Optimization.

### 2.1 PRELIMINARIES OF PROTEINS

A protein is composed of an amino acid sequence that folds into 3D structures. We denote a protein with a residue sequence of length $n$ by $\mathcal{P} = (\mathcal{S}, \mathcal{C})$, where $\mathcal{S}$ is the set of sequential residues and $\mathcal{C}$ denotes the spatial coordinates of residues. Specifically, the residue sequence is defined as $\mathcal{S} = [a_1, a_2, \ldots, a_n]$, with each residue $a_i \in \mathcal{A}$, where $\mathcal{A}$ denotes the set of 20 standard amino acids. The residue coordinates are given as $\mathcal{C} = [c_1, c_2, \ldots, c_n]$, where each $c_i = [x_i, y_i, z_i]^\top \in \mathbb{R}^3$ represents the 3D coordinate of the C$\alpha$ atom [1] corresponding to residue $a_i$. More information about the sequence and structures of the proteins can be found in Appendix C.

### 2.2 PRETRAINING TASKS AND PRETRAINING MODELS

Recently, the pretraining paradigm has achieved remarkable success across text, images, and graphs (Devlin et al., 2019; Achiam et al., 2023; Radford et al., 2021; Hu et al., 2020). To facilitate protein modeling, various pretraining strategies and model architectures have also been investigated (Rives et al., 2021; Lin et al., 2023; Brandes et al., 2022; Su et al., 2024). In this work, our Protap conducts a comprehensive analysis of representative protein pretraining tasks and models to systematically understand their capabilities and limitations in downstream applications. The models and pretraining tasks provided in Protap are summarized in Tab. 2 and briefly introduced below.

---

[1]The C$\alpha$ atom serves as a stable backbone reference for each residue, commonly used due to its consistency and significance in protein structure modeling.

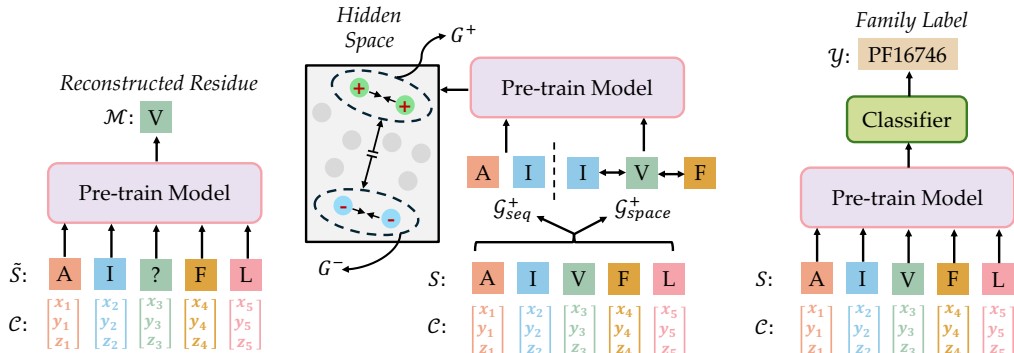

(a) Illustration of pretraining tasks in our Protap. (I) **Masked Language Modeling** is a self-supervised objective designed to recover masked residues in protein sequences; (II) **Multi-View Contrastive Learning** leverages protein structural information by aligning representations of biologically correlated substructures. Given two views of the same protein; (III) **Protein Family Prediction** introduces functional and structural supervision by training models to predict family labels based on protein sequences and 3D structures.

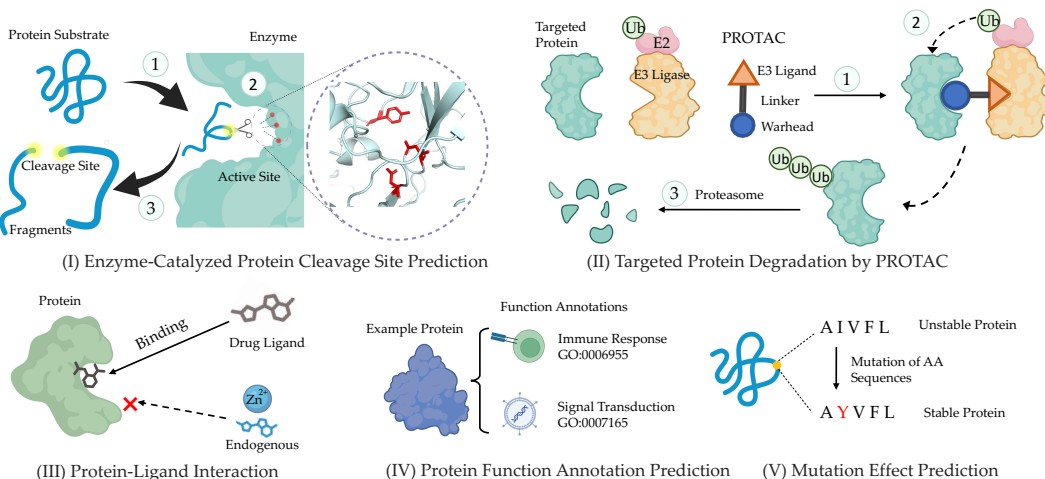

(b) Illustration of downstream applications in our Protap. (I) Biological process of the **enzyme-catalyzed protein hydrolysis**; (II) Biological process of **targeted protein degradation by PROTACs**, where PROTACs form a ternary complex with the target protein and E3 ligase, leading to target protein ubiquitination and degradation; (III) In **protein-ligand interaction**, the ligand binds to the protein pocket, blocking interactions with other molecules; (IV) **Protein function annotation prediction** reveals the biological activities a protein participates in; (V) **Mutation effect prediction** optimizes protein properties or functions for protein engineering.

Figure 1: Overview of pre-training and downstream tasks in Protap.

**Pretraining Task 1: Masked Language Modeling (MLM).** This pretext task is based on the sequence information. A protein's amino acid sequence adheres to an inherent grammar that encodes its structural and functional properties (Anfinsen, 1973). Therefore, inspired by the success of text modeling, masked language modeling has been adopted to train the protein language model (Rives et al., 2021) to capture patterns underlying residue sequences. Specifically, as Fig. 1a (a) shows, MLM aims to fill in missing amino acids in protein sequences given the masked residue sequence of a protein. ProteinBERT (Brandes et al., 2022), ESM2 (Lin et al., 2023), ESM Carbrain (Hayes et al., 2025) and various other protein language models have been pretrained using this MLM task.

**Pretraining Task 2: Multi-View Contrastive Learning (MVCL).** This pretext task is based on the structural information. Protein structure plays a vital role in determining its biological function. Furthermore, the local structures (motifs) within a protein are biologically related (Ponting & Russell, 2002; Mackenzie & Grigoryan, 2017). Inspired by this, multi-view contrastive learning (MVCL) has been extended to preserve the representation similarity between the correlated substructures of

proteins (Zhang et al., 2023). As shown in Fig. 1a (b), given a protein $\mathcal{P}$, we generate two views $\mathcal{G}_{seq}^{+}$ and $\mathcal{G}_{space}^{+}$ from its amino acid sequence as positive samples. Views from unrelated proteins serve as negative samples. The MVCL objective is to align positive samples while contrasting them with negative samples in the hidden representation space.

**Pretraining Task 3: Protein Family Prediction (PFP).** Different from the MLM and MVLC, this task leverages auxiliary protein knowledge to introduce functional and structural supervision into protein representation learning. A protein family is a group of evolutionarily related proteins descended from a common ancestor, typically sharing similar three-dimensional structures, functions, and significant sequence similarities (Sillitoe et al., 2021; Bateman et al., 2004). Consequently, the Protein Family Prediction (PFP) task has been utilized to learn structurally contextualized representations (Min et al., 2021). As illustrated in Fig. 1a (c), Protap also deploys the protein family prediction to predict the ground-truth family labels $y$ given the sequence $\mathcal{S}$ and the coordinate $\mathcal{C}$.

**Pretraining Models**. In Protap, we incorporate and evaluate the following three categories of model architectures pretrained by the aforementioned pretext tasks: (**i**) Sequence-based model: Protein-BERT (Brandes et al., 2022), ESM-2 (Lin et al., 2023) and ESM Carbrain (Hayes et al., 2025) treat the protein sequences as biological languages and adopt the transformer (Vaswani et al., 2017) to extract the sequential patterns. (**ii**) Structure-based model: Protap also implementes EGNN (Satorras et al., 2021), SE(3) transformer (Fuchs et al., 2020), and GVP (Jing et al., 2021), which are representative equivariant architectures for structural information encoding. (**iii**) Sequence-structure hybrid model: SaProt (Su et al., 2024) enhances the protein language model with explicit structure vocabulary. Additionally, inspired by Transformer-M (Luo et al., 2023a) and Unizyme (Li et al., 2025), Protap implements a D-Transformer, which introduces the residue distance matrix into the sequence attention computation as shown in Tab. 2. Protap pretrains ProteinBERT, EGNN, SE3 transformer, GVP, and D-Transformer on each aforementioned pretraining task with UniProt (Consortium, 2019), yielding 15 pretraining models. For the ESM-2 and SaProt, we directly adopt their publicly available pretrained weights for protein modeling. More details of pretraining model architectures, pretraining datasets, and other pretraining details are given in Appendix F.

## 2.3 APPLICATIONS AND DOMAIN MODELS

In this subsection, we introduce the applications adopted in Protap, categorized into specialized and general. Specialized applications focus on specific proteins or biological processes, while general applications broadly apply across diverse proteins. We briefly describe each application along with representative domain models. A complete list of implemented domain models is provided in Tab. 2, and detailed experimental setups including datasets and metrics are available in Appendix D.

**Specialized Application 1: Enzyme-Catalyzed Protein Cleavage Site Prediction (PCS).** As illustrated in Fig. 1b, proteolytic enzymes will first recognize specific amino acid sequences or structural motifs within substrate proteins with the enzyme's active sites. Then, the enzymes catalyze the cleavage of peptide bonds at the cleavage site. This fundamental biological process regulates protein activity, turnover, and signaling across diverse biological contexts. Predicting the protein cleavage sites under the catalysis of enzymes has many crucial applications, such as enzyme engineering and therapeutic target identification. For example, in the design of enzyme inhibitors or prodrugs, identifying key cleavage peptides under the catalysis of the HIV enzyme could enhance drug specificity (Lv et al., 2015). The enzyme-catalyzed protein cleavage site prediction can be formulated as a residue-level binary classification task. Specifically, let $\mathcal{P}^s$ and $\mathcal{P}^e$ denote the protein substrate and enzyme. The cleavage site predictor aims to learn the following function:

$$f : (\mathcal{P}^s, \mathcal{P}^e) \rightarrow \{0, 1\}^{|\mathcal{P}^s|}. \tag{1}$$

The majority of existing methods, such as Procleave (Li et al., 2020a), construct enzyme-specific features to identify the cleavage sites under the enzyme of interest. Recently, UniZyme (Li et al., 2025) has leveraged both enzyme and substrate encoders to build a unified cleavage site predictor that generalizes across various enzymes.

**Specialized Application 2: Targeted Protein Degradation by Proteolysis-Targeting Chimeras (PROTACs).** A Proteolysis-targeting chimeras (PROTAC) is a heterobifunctional molecule consisting of three components: a ligand for the targeted protein (commonly referred to as the warhead), a chemical linker, and a ligand that recruits an E3 ubiquitin ligase. PROTACs have emerged as powerful tools for selectively degrading disease-associated proteins via the ubiquitin-proteasome

system (Marei et al., 2022). As illustrated in Fig. 1b, the PROTAC-mediated degradation process begins with simultaneous binding of the PROTAC to the target protein and an E3 ligase, resulting in a ternary complex. This complex subsequently facilitates the transfer of ubiquitin from the E2 enzyme to the target protein, ultimately leading to its proteasomal degradation. PROTACs offer several advantages over conventional treatments. First, due to their catalytic mode of action, PROTACs remain effective at lower doses, potentially reducing side effects (Neklesa et al., 2017; Toure & Crews, 2016). Second, unlike traditional drugs that rely on accessible binding pockets, PROTACs can target proteins previously considered undruggable and overcome resistance caused by mutations near active sites (Garber, 2022; Edmondson et al., 2019). However, the flexible ternary structure of PRO-TACs also introduces a more complex structure-activity relationship, making modeling this process more challenging. Formally, the prediction of PROTAC-mediated degradation of a target protein $\mathcal{P}^t$ by an E3 ligase can be defined as:

$$f : (\underbrace{\text{Warhead, Linker, E3 ligand}}_{\text{PROTAC}}, \text{ E3 ligase, } \mathcal{P}^t) \rightarrow \{0, 1\} \qquad (2)$$

 To achieve the above task, initial efforts have been conducted to model the ternary complex formation induced by PROTAC molecules (Li et al., 2022; Cai et al., 2025): DeepPROTACs employs GCNs to encode the ternary complex. ET-PROTAC considers the cross-talk between the PROTAC, target protein, and E3 ligase, which enables the modeling of the complex process in the targeted protein degradation.

**General Application 3: Protein–Ligand Interactions (PLI).** As illustrated in Fig. 1b, protein–ligand binding refers to the highly specific and affinity-driven interaction between a protein and a small molecule ligand, resulting in a stable complex that can block the binding of other molecules (Miller & Dill, 1997). The binding affinity can be quantified by the Gibbs free energy change $\Delta G$ of the protein-ligand complex, where a more negative $\Delta G$ indicates stronger binding affinity. Accurate prediction of protein–ligand binding affinity enables efficient identification of promising drug candidates and accelerates the optimization of therapeutic molecules. For example, kinases play critical roles in regulating cell growth and survival pathways implicated in cancer. Thus, predicting kinase–ligand binding affinity facilitates the effective screening of kinase inhibitors, which can selectively block these pathways. The screened kinase inhibitors are promising candidates for targeted cancer therapy. Formally, predicting protein–ligand binding affinity can be formulated as a regression task: $f : (\mathcal{P}, \mathcal{M}) \rightarrow \mathbb{R}$, where $\mathcal{P}$ denotes the protein and $\mathcal{M}$ denotes the ligand molecule. Some domain-specific methods, such as MONN (Li et al., 2020b), do not rely on structural information. To mitigate the false positive issue, KDBNet (Luo et al., 2023b) applied an uncertainty recalibration technique to refine the uncertainty estimates.

**General Application 4: Protein Function Annotation Prediction (PFA)**. To systematically represent protein functions, the Gene Ontology (GO) annotation framework was developed (Ashburner et al., 2000), providing structured annotations across Molecular Function (MF), Cellular Component (CC), and Biological Process (BP). For instance, in the Molecular Function (MF) aspect, a protein may be annotated with specific biochemical activities, such as immune Response or signal transduction. Among the known protein sequences, fewer than 1% have experimentally validated functional annotations. Predicting the GO terms of proteins helps researchers better understand protein functions and potentially guide the discovery and design of proteins with desired functional properties. Protein function prediction is inherently a multi-label classification task. Early methods relied on sequence alignment or direct modeling of the sequence. More recent approaches have progressively integrated additional modalities. For example, DeepFRI (Gligorijević et al., 2021) combines protein structure and pre-trained sequence embeddings using a GCN, DPFunc (Wang et al., 2025) further incorporates domain-level information, and the PLM-based DeepGO-SE (Kulmanov et al., 2024).

**General Application 5: Mutation Effect Prediction for Protein Optimization (MTP).** Protein mutations refer to the substitution, insertion, or deletion of amino acids within protein sequences. As illustrated in Fig. 1b, mutations are frequently associated with changes in functional properties such as stability, binding affinity, and pathogenicity. Computational models can explore the protein fitness landscape to map sequences or structures to functional properties, which can enable protein optimization with mutations. For example, mutations in antibodies or protein complexes can be systematically explored to discover variants with enhanced binding affinity (Cai et al., 2024). MTP can be formulated as estimating changes in target properties (e.g., stability changes measured by $\Delta\Delta G$) resulting from protein mutations. Due to limited experimental annotations, mutation effect

prediction typically relies on a zero-shot learning setting. Recently, protein language models (PLMs) have shown promise in mutation effect prediction due to their ability to capture patterns learned from evolutionary data. PLMs tend to assign higher probabilities to mutations that are consistent with evolutionary patterns, and these mutations are more likely to yield beneficial functional effects. Details of the zero-shot mutation effective prediction with PLMs can be found in Appendix D.5.

## 3 EXPERIMENTS

In this section, we conduct empirical studies to compare domain models and various pretraining frameworks and strategies on the downstream applications described in Sec. 2.3.

### 3.1 EXPERIMENTAL SETUP

**Pre-training Dataset**. For pre-training, we collect 542,378 protein structures from AlphaFold Protein Structure Database (AFDB) and obtain the corresponding amino acid sequences and protein family labels from UniProt (Varadi et al., 2022; Consortium, 2019). This curated dataset spans diverse and non-redundant protein structures, enabling a comprehensive evaluation of different training strategies. ESM-2 and SaProt are trained on protein sequences from UniRef (Consortium, 2019), while ESM-C further incorporates sequences from MGnify (Richardson et al., 2023) and the JGI (Nordberg et al., 2014). Additional dataset details are provided in Appendix F.2.

**Downstream Application Dataset.** Each downstream task uses the same dataset for both training-from-scratch and fine-tuning to ensure fair comparison. All datasets are collected from peer-reviewed sources and standardized through quality control and formatting, as detailed in Appendix D. Specifically, PROTACs data are from PROTAC-DB (Ge et al., 2025), cleavage data from MEROPS (Rawlings et al., 2014), protein–ligand interactions from KDBNet, function prediction from DeepFRI (Gligorijević et al., 2021) (with structural data from ProteinWorkshop (Jamasb et al., 2024)), and mutation prediction from ProteinGym (Notin et al., 2023).

**Downstream Task Training.** Pretrained models serve as protein encoders in two settings: (i) *Training from scratch*, where all parameters are randomly initialized and trained end-to-end; (ii) *Freeze encoder fine-tuning*, where encoders are initialized with pretrained weights and frozen, while only task-specific heads are updated; (iii) *Full parameter fine-tuning*, where both encoders and task-specific heads are updated. All domain-specific models are trained from scratch.

**Molecular Encoders in Downstream Applications**. The applications of PLI and PROTACs involve various chemical molecules in the interactions with proteins, requiring a molecular encoder. To maintain consistency across interaction tasks, molecular components, drugs in PLI, warheads, linkers, and E3-ligands in PROTACs are encoded using randomly initialized GVP encoders. For domain-specific models, molecular encoders follow their original implementations.

**Model Setups**. We use fixed random seeds $42, 128, 256, 512, 1024$ for reproducibility, and report mean $\pm$ standard deviation over five runs. All models are trained with the Adam optimizer. The learning rate follows a linear warm-up over 10% of training steps, then cosine annealing to 1e-6, with a peak of 1e-4. More training details are provided in Appendix D.

### 3.2 IMPACTS OF VARIOUS PRETRAINING ON DOWNSTREAM PERFORMANCE

Our experiments aim to make comparisons across the following three perspectives: (i) *The scalability of pretraining* (e.g., ESM-2 vs. compact models like EGNN). (ii) *The model architectures* (e.g., structural models vs. protein language models). (iii) *The strategies of pretraining*. Therefore, we examine three pretraining tasks employed in Protap as described in Sec. 2.2, which are MLM, MVCL, and PFP. We pre-train EGNN, SE(3) Transformer (SE(3) Trans), GVP, ProteiBERT (ProtBERT), and D-Transformer (D-Trans) with all three tasks with 542k proteins to obtain pretraining encoders, which are listed in Tab. 2 and are detailed in Appendix F. For ESM-2 and ESM-C, we testify to the publicly released pretrained weights that are learned via MLM training on 70M proteins.

Here, we discuss on the training from scratch and the freeze encoder fine-tuning. The results are in Tab. 3, and we answer the following research questions. Results under full parameter finetuning are provided in Appendix A.2.

**RQ1: Can frozen pretraining encoders outperform a supervised encoder that trained on the downstream training set from scratch?**

Table 3: Performance comparison across model architectures under different training strategies. The first line for each model (e.g., EGNN) denotes a randomly initialized protein encoder trained purely with downstream task supervision. The subsequent lines (e.g., w/MLM) represent pretrained encoders with frozen weights, where only the task-specific head is fine-tuned. Results under full parameter finetuning are provided in Appendix A.2. Mutation effect prediction (MTP) is only applicable to masked language modeling.

| Model | PCS | | | | PROTACs | | PLI | | PFA | | MTP | |
|---|---|---|---|---|---|---|---|---|---|---|---|---|
| | C14.005 | | M10.003 | | PROTACDB | | Davis | | MF | | DMS | |
| | AUC(%)↑ | AUPR(%)↑ | AUC(%)↑ | AUPR(%)↑ | Acc(%)↑ | AUC(%)↑ | MSE↓ | Pear(%)↑ | Fmax(%)↑ | AUPR(%)↑ | Pear(%)↑ | AUC(%)↑ |
| EGNN | 95.51 ± 0.03 | 35.44 ± 0.06 | 90.24 ± 0.32 | 15.04 ± 1.18 | 80.19 ± 1.23 | 88.55 ± 1.78 | 0.492 ± 0.031 | 57.11 ± 1.11 | 4.03 ± 0.18 | 4.75 ± 0.08 | – | – |
| w/MLM | 95.11 ± 0.31 | 19.61 ± 1.16 | 79.19 ± 0.11 | 4.12 ± 0.12 | 80.24 ± 1.20 | 88.19 ± 1.59 | 0.486 ± 0.058 | 56.56 ± 1.47 | 4.21 ± 0.01 | 6.52 ± 0.05 | 30.25 | 65.68 |
| w/MVCL | 94.80 ± 0.00 | 19.61 ± 1.16 | 82.84 ± 0.68 | 3.23 ± 0.13 | 78.78 ± 2.57 | 86.02 ± 1.00 | 0.510 ± 0.035 | 52.68 ± 5.15 | 4.13 ± 0.01 | 5.78 ± 0.00 | – | – |
| w/PFP | 95.09 ± 0.41 | 27.95 ± 0.99 | 82.45 ± 0.73 | 5.43 ± 0.87 | 79.72 ± 1.44 | 87.64 ± 1.01 | 0.483 ± 0.013 | 55.74 ± 1.92 | 9.06 ± 0.18 | 8.23 ± 0.09 | – | – |
| SE(3) Trans | 81.76 ± 3.11 | 2.07 ± 0.81 | 67.12 ± 0.05 | 1.01 ± 0.01 | 79.23 ± 1.03 | 87.59 ± 1.55 | 0.566 ± 0.137 | 55.91 ± 1.24 | 3.93 ± 0.00 | 4.10 ± 0.23 | – | – |
| w/MLM | 67.65 ± 1.31 | 0.55 ± 0.03 | 66.81 ± 0.54 | 1.15 ± 0.11 | 80.56 ± 2.08 | 87.64 ± 1.45 | 0.536 ± 0.033 | 55.83 ± 1.51 | 3.42 ± 0.00 | 2.37 ± 0.03 | 20.63 | 59.87 |
| w/MVCL | 44.01 ± 0.72 | 0.20 ± 0.00 | 52.16 ± 2.50 | 0.53 ± 0.07 | 68.11 ± 5.57 | 77.84 ± 2.23 | 0.819 ± 0.164 | 48.42 ± 2.86 | 3.39 ± 0.00 | 2.38 ± 0.01 | – | – |
| w/PFP | 69.27 ± 2.55 | 0.81 ± 0.17 | 66.24 ± 3.13 | 1.20 ± 0.16 | 79.24 ± 1.16 | 87.53 ± 1.07 | 0.539 ± 0.038 | 52.06 ± 2.30 | 3.55 ± 0.06 | 2.58 ± 0.10 | – | – |
| GVP | 95.55 ± 0.46 | 17.99 ± 5.02 | 86.05 ± 1.68 | 5.40 ± 0.52 | 70.67 ± 1.42 | 77.58 ± 1.12 | 0.516 ± 0.007 | 49.52 ± 2.59 | 5.41 ± 0.11 | 2.15 ± 0.03 | – | – |
| w/MLM | 93.23 ± 0.34 | 8.31 ± 0.87 | 80.94 ± 0.45 | 4.03 ± 0.11 | 70.03 ± 2.16 | 77.51 ± 1.07 | 0.524 ± 0.007 | 48.52 ± 2.62 | 9.21 ± 0.26 | 4.05 ± 0.12 | 20.78 | 60.65 |
| w/MVCL | 94.84 ± 0.25 | 8.74 ± 0.66 | 83.97 ± 0.27 | 3.69 ± 0.19 | 70.75 ± 2.15 | 78.45 ± 1.30 | 0.505 ± 0.006 | 51.15 ± 2.00 | 9.54 ± 0.59 | 4.89 ± 0.29 | – | – |
| w/PFP | 62.12 ± 0.43 | 0.32 ± 0.01 | 57.60 ± 0.73 | 0.61 ± 0.02 | 70.83 ± 1.32 | 77.64 ± 1.32 | 0.518 ± 0.006 | 49.24 ± 1.15 | 10.12 ± 0.07 | 5.30 ± 0.02 | – | – |
| ProtBERT | 95.49 ± 0.09 | 40.49 ± 0.25 | 89.83 ± 0.05 | 10.74 ± 0.12 | 75.38 ± 1.51 | 84.13 ± 1.89 | 0.521 ± 0.020 | 50.23 ± 1.24 | 3.39 ± 0.00 | 2.93 ± 0.01 | – | – |
| w/MLM | 95.45 ± 0.10 | 9.37 ± 0.62 | 85.56 ± 0.76 | 7.12 ± 0.06 | 78.20 ± 1.14 | 86.75 ± 1.78 | 0.519 ± 0.033 | 52.67 ± 1.13 | 4.15 ± 0.01 | 4.27 ± 0.01 | 14.20 | 60.92 |
| w/MVCL | 95.78 ± 0.21 | 28.51 ± 0.06 | 86.13 ± 0.19 | 7.16 ± 0.39 | 79.06 ± 2.71 | 86.45 ± 1.08 | 0.553 ± 0.049 | 55.82 ± 1.67 | 3.51 ± 0.00 | 4.66 ± 0.03 | – | – |
| w/PFP | 64.25 ± 0.35 | 0.36 ± 0.00 | 57.29 ± 0.10 | 0.67 ± 0.01 | 80.51 ± 1.03 | 87.46 ± 1.71 | 0.533 ± 0.048 | 54.81 ± 2.08 | 6.05 ± 0.05 | 5.64 ± 0.01 | – | – |
| D-Trans | 97.60 ± 0.08 | 61.42 ± 1.30 | 88.28 ± 0.79 | 22.82 ± 2.95 | 80.08 ± 1.08 | 86.58 ± 0.28 | 0.416 ± 1.028 | 60.92 ± 0.22 | 19.57 ± 0.80 | 11.16 ± 0.67 | – | – |
| w/MLM | 96.53 ± 0.52 | 20.12 ± 2.86 | 84.10 ± 0.66 | 11.73 ± 1.41 | 77.78 ± 1.93 | 85.16 ± 0.89 | 0.494 ± 1.116 | 55.79 ± 0.50 | 17.92 ± 0.14 | 9.85 ± 0.09 | -0.09 | 56.84 |
| w/MVCL | 95.23 ± 0.46 | 41.26 ± 10.3 | 83.50 ± 0.79 | 15.85 ± 1.30 | 74.16 ± 1.44 | 82.32 ± 0.81 | 0.550 ± 2.620 | 54.21 ± 0.08 | 15.87 ± 0.26 | 8.00 ± 0.20 | – | – |
| w/PFP | 96.40 ± 0.59 | 39.56 ± 5.55 | 85.51 ± 0.42 | 12.29 ± 1.26 | 74.76 ± 1.08 | 82.34 ± 0.09 | 0.445 ± 1.064 | 56.46 ± 0.01 | 17.46 ± 0.18 | 9.46 ± 0.12 | – | – |
| ESM-2 | 97.23 ± 0.06 | 41.22 ± 0.25 | 86.34 ± 0.12 | 6.16 ± 0.07 | 78.46 ± 3.03 | 85.74 ± 3.47 | 0.491 ± 4.823 | 53.25 ± 1.26 | 49.79 ± 0.03 | 43.44 ± 0.17 | 43.05 | 73.48 |
| ESM-C | 95.76 ± 0.01 | 38.28 ± 0.08 | 88.08 ± 0.00 | 8.06 ± 0.08 | 80.47 ± 0.90 | 86.74 ± 0.37 | 0.461 ± 0.037 | 53.41 ± 0.87 | 38.27 ± 0.16 | 30.31 ± 0.12 | 42.51 | 73.24 |
| SaProt | 96.16 ± 0.16 | 42.91 ± 0.27 | 89.18 ± 0.06 | 13.48 ± 0.03 | 73.44 ± 1.32 | 80.29 ± 0.70 | 0.473 ± 0.004 | 51.66 ± 0.67 | 33.06 ± 0.09 | 25.21 ± 0.11 | 49.22 | 76.89 |

Overall, supervised encoders trained from scratch tend to outperform the pretraining encoders even when they are trained with large-scale datasets. For example, though ESM-2 (650M parameters) already achieves competitive performance, ESM-2 is often worse than the EGNN (10M parameters) encoder trained for the downstream tasks from scratch. This suggests a degree of mismatch between the pretraining objectives and the downstream tasks, indicating that training from scratch enables the model to learn task-specific representations that are more aligned with the requirements of the downstream task.

**RQ2: In what ways do variations in pretraining objectives impact the effectiveness of models on downstream tasks?**

No significant patterns were observed regarding the impact of different pretraining strategies on downstream tasks. Various pretraining approaches do not exhibit a clear preference for specific downstream tasks. However, excluding models trained from scratch, those pretrained with PFP consistently achieve the best results on the PFA task compared to other pretraining methods, supporting the claim that proteins within the same family often share similar biological functions, and incorporating protein family information during pretraining aids functional prediction.

**RQ3: Are models incorporating structures better than those purely using protein sequences?**

Models that incorporate structural information generally achieve superior performance. Except for the PFA task, models such as EGNN, SE(3)-Trans, and D-Trans, which integrate spatial features, consistently outperform larger pretrained protein language models such as ESM-2 and ProtBERT. This observation highlights two key insights: (i) The inclusion of three-dimensional structural information provides essential inductive biases that are absent in sequence-only architectures. (ii) Architectural alignment with biochemical properties can be more important than the sheer size of pretrained models in achieving strong performance on downstream protein-related tasks.

## 3.3 COMPARISON BETWEEN PRETRAINED MODELS AND DOMAIN-SPECIFIC MODELS

We compare the pretraining model architectures with eight domain-specific models, which are listed in Tab. 2. Since the representative solution of mutation effect prediction is protein language models, we omit the comparison on mutation effect prediction. For the pretraining model architectures, we

Table 4: Comparisons between general and domain models across four downstream tasks.

(a) Comparisons between general and domain models on protein cleavage site prediction.

| | Metrics | General Architecture | | | | | Domain Model | |
|---|---|---|---|---|---|---|---|---|
| | | EGNN | SE3 | GVP | ProtBERT | D-Transformer | Clipzyme | Unizyme |
| C14.005 | AUC(%)↑ | 95.51 ± 0.03 | 81.76 ± 3.11 | 95.55 ± 0.46 | 95.49 ± 0.09 | 97.60 ± 0.08 | 92.27 ± 0.40 | 96.23 ± 0.10 |
| | AUPR(%)↑ | 35.44 ± 0.06 | 2.07 ± 0.81 | 17.99 ± 5.02 | 40.49 ± 0.25 | 61.42 ± 1.30 | 43.23 ± 1.00 | 52.20 ± 0.90 |
| M10.003 | AUC(%)↑ | 90.24 ± 0.32 | 67.12 ± 0.05 | 86.05 ± 1.68 | 89.83 ± 0.05 | 88.28 ± 0.79 | 82.50 ± 0.30 | 87.04 ± 0.20 |
| | AUPR(%)↑ | 15.04 ± 1.18 | 1.01 ± 0.01 | 5.40 ± 0.52 | 10.74 ± 0.12 | 22.82 ± 2.95 | 5.81 ± 0.20 | 7.28 ± 0.40 |

(b) Comparisons between general and domain models on PROTACs.

| Metrics | General Architecture | | | | | Domain Model | |
|---|---|---|---|---|---|---|---|
| | EGNN | SE3 w/MLM | GVP w/MVCL | ProtBERT w/PFP | D-Transformer w/PFP | DeepPROTACs | ET-PROTACs |
| Acc(%)↑ | 80.19 ± 1.23 | 80.56 ± 2.08 | 70.75 ± 2.15 | 80.51 ± 1.03 | 80.08 ± 1.08 | 70.12 ± 0.83 | 78.87 ± 0.88 |
| AUC(%)↑ | 88.55 ± 1.78 | 87.64 ± 1.45 | 78.45 ± 1.30 | 87.46 ± 1.71 | 86.58 ± 0.28 | 77.59 ± 1.22 | 78.64 ± 1.21 |

(c) Comparisons between general and domain models on Protein-Ligand Interaction.

| Metrics | General Architecture | | | | | Domain Model | |
|---|---|---|---|---|---|---|---|
| | EGNN | SE3 | GVP w/MVCL | ProtBERT w/MVCL | D-Transformer | KDBNet | MONN |
| MSE↓ | 0.492 ± 0.031 | 0.566 ± 0.137 | 0.505 ± 0.006 | 0.553 ± 0.049 | 0.416 ± 1.02 | 0.342 ± 0.018 | 0.750 ± 0.054 |
| Pear(%)↑ | 57.11 ± 1.11 | 55.91 ± 1.24 | 51.15 ± 2.00 | 55.82 ± 1.67 | 60.92 ± 0.22 | 70.97 ± 1.82 | 50.30 ± 3.60 |

(d) Comparisons between general and domain models on protein function prediction.

| Metrics | General Architecture | | | | | Domain Model | |
|---|---|---|---|---|---|---|---|
| | EGNN w/PFP | SE3 | GVP w/PFP | ProtBERT w/PFP | D-Transformer | DeepFRI | Deepfunc |
| Fmax(%)↑ | 9.06 ± 0.18 | 3.93 ± 0.00 | 10.12 ± 0.07 | 6.05 ± 0.05 | 19.57 ± 0.80 | 26.01 ± 0.45 | 45.40 ± 0.41 |
| AUPR(%)↑ | 8.23 ± 0.09 | 4.10 ± 0.23 | 5.30 ± 0.02 | 5.64 ± 0.01 | 11.16 ± 0.67 | 32.77 ± 0.58 | 50.84 ± 0.30 |

report the best training strategies according to Tab. 3. The comparison between pretrained model architectures and domain-specific architectures is given in Tab. 4 (a)-(d). In particular, we aim to answer the following research questions.

**RQ4: How do the domain-specific models perform compared with general ones?**

The performance of domain-specific models and pretrained models varies across different tasks. We observe that on the protein-ligand interaction and protein function annotation prediction task, the domain-specific models could outperform the general model architectures by a large margin. However, for the applications of PROTACs that exhibit complex interaction processes, the general framework EGNN exhibits great results. SaProt greatly outperforms other models in mutation effects prediction, including ESM-2. This is likely attributable to the Structure-Vocabulary, which reduces local sequence ambiguity and mitigates semantic confusion.

**RQ5: To what extent does incorporating biochemical inductive biases improve generalization across Enzyme-Catalyzed Protein Cleavage Site Prediction?**

EGNN, UniZyme, and D-Trans exhibit superior performance on the protein task, e.g., D-Trans achieves $97.60 \pm 0.08$ on C14.005 while EGNN achieves $90.24 \pm 0.32$ on M10.003. UniZyme obtains comparable results. These indicate that the use of biochemical priors, such as energy frustration and distance matrix, facilitates improved detection of functional regions in proteins.

## 4 CONCLUSION AND FUTURE WORKS

We introduce Protap, a unified benchmark that brings together general-purpose pretraining architectures and domain-specific models to evaluate five key protein modeling applications under a standard, reproducible framework. Our extensive experiments indicate that no single model or pretraining objective can outperform all others across applications. The architecture and training strategy must be chosen based on the specific characteristics of each task. There are two directions that need further investigation: (i) We will further explore the scaling laws that govern how model capacity and pretraining data volume translate into downstream gains. (ii) We will extend Protap to holistically cover protein design tasks, e.g., peptide design, enzyme design.

## 5 REPRODUCIBILITY STATEMENT

Our implementations are based on widely adopted architectures, pretraining strategies, and downstream evaluation pipelines for protein modeling. The code is available at `https://anonymous.4open.science/r/protap-1CC5`. To ensure reproducibility, we provide detailed descriptions of the pretraining objectives in Section 2.2, along with the corresponding training configurations in Appendix F.2. Hyperparameters for all models, including learning rate schedules, batch sizes, and optimization settings, are explicitly reported in Section 3 and Appendix F.2. The datasets used for both pretraining and downstream evaluation are publicly available at `https://doi.org/10.5281/zenodo.17192817`, and we specify their sources in Tab. 2 (b), and provide more details in Appendix D. The exact evaluation metrics are specified in Section Appendix D. These details, combined with the systematic comparison across architectures and strategies presented in Tab. 3 and Tab. 4, allow independent researchers to reproduce our results based on the manuscript alone.

## 6 ETHICS STATEMENT

We introduce Protap, a benchmark designed to systematically evaluate protein models on diverse downstream tasks without additional training. Protap itself does not generate or modify proteins; rather, it provides a standardized platform for assessing model capabilities. As such, the ethical implications primarily depend on how insights from Protap are applied. Potentially beneficial applications include accelerating biomedical research, enabling more reliable drug discovery pipelines, and supporting the development of sustainable biotechnologies. While there is a theoretical risk that advances in model evaluation could indirectly inform harmful misuse, such risks are significantly outweighed by the anticipated societal benefits of improving our understanding of protein modeling and ensuring the responsible development of computational biology tools.

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

APPENDIX CONTENTS

## A Supplemental Experiments and Observations

### A.1 Descriptions and Categorization of Domain-Specific Models

The primary distinction between domain models and general models lies in their task-specific design. Domain models are tailored for particular tasks by enabling interaction between representations of different components and incorporating biochemical priors to enhance the encoder. In addition, domain models also leverage knowledge from other biochemical databases to preprocess the data, enabling the extraction of task-relevant key regions. Below, we summarize the characteristics of each domain model in relation to its respective tasks. Detailed task descriptions and model architectures are found in Appendix D and Appendix E.

Table 5: Domain Model Categorization

| Category | Model |
| --- | --- |
| Biochemical Prior Enhancement | UniZyme, CLIPZyme, KDBNet, DeepFRI |
| Cross-Component Interaction | ET-PROTACs, DeepPROTACs |
| Hybrid | DPFunc, MONN |

**Enzyme-Catalyzed Protein Cleavage Site Prediction (PCS)**

- **UniZyme**. UniZyme (Li et al., 2025) not only incorporates pretraining for enzyme active site prediction but also introduces energetic frustration, an intrinsic biophysical phenomenon that directly reflects active site properties and catalytic mechanisms. The integration of such biochemical priors enhances the generalization capability of the enzyme encoder.

- **ClipZyme**. ClipZyme (Mikhael et al., 2024) formulates enzyme function prediction as a reaction-centric retrieval task, aligning enzyme representations with chemical reaction embeddings in a shared latent space. By explicitly modeling atom-mapped reaction graphs and constructing pseudo-transition states, the framework integrates reaction mechanism information that is specific to enzymatic catalysis.

**Targeted Protein Degradation by Proteolysis-Targeting Chimeras (PRAOTACs)**

- **DeepPROTACs**. DeepPROTACs (Li et al., 2022) circumvent explicit modeling of the ternary complex by encoding different components of the Target protein–PROTAC–E3 ligase system using separate neural network modules.

- **ET-PROTACs**. ET-PROTACs (Cai et al., 2025) utilizes a cross-modal strategy and ternary attention mechanism, the model fully accounts for the cross-talk between PROTACs, target proteins, and E3 ligases, enabling more accurate modeling of ternary complex interactions.

**Protein–Ligand Interactions (PLI)**

- **KDBNet**. KDBNet (Luo et al., 2023b) does not model the entire protein, instead, it defines the binding pocket based on prior knowledge from the KLIFS database. In modeling kinases, it incorporates both geometric and evolutionary features, including backbone torsion angles and embeddings derived from the ESM language model.

- **MONN**. MONN (Li et al., 2020b) is a multi-objective neural network designed to simultaneously predict non-covalent interactions and binding affinity between compounds and proteins, without relying on structural information.

**Protein Function Annotation Prediction (PFA)**

- **DeepFRI**. DeepFRI (Gligorijević et al., 2021) employs graph convolutional networks (GCNs) (Kipf & Welling, 2017) to process protein contact maps and integrates representations pretrained on Pfam sequences.

- **DPFunc**. DPFunc (Wang et al., 2025) identifies key functional regions within protein structures and precisely predicts their associated biological functions by leveraging protein domain information.

## A.2 ADDITIONAL OBSERVATIONS

Table 6: Performance comparison across model architectures under different training strategies with full parameter fine-tuning.

| Model | PCS | | PROTACs | | PLI | |
|---|---|---|---|---|---|---|
| | C14.005 | | PROTACDB | | Davis | |
| | AUC(%)↑ | AUPR(%)↑ | Acc(%)↑ | AUC(%)↑ | MSE↓ | Pear(%)↑ |
| EGNN w/MLM | 95.36 ± 0.06 | 19.23 ± 0.31 | 79.03 ± 0.61 | 88.50 ± 0.37 | 0.461 ± 0.010 | 57.56 ± 1.18 |
| EGNN w/MVCL | 94.80 ± 0.00 | 19.61 ± 1.16 | 76.89 ± 2.50 | 88.81 ± 0.50 | 0.484 ± 0.031 | 56.18 ± 0.12 |
| EGNN w/PFP | 95.09 ± 0.41 | 27.95 ± 0.99 | 79.01 ± 0.55 | 87.65 ± 0.33 | 0.523 ± 0.036 | 55.44 ± 2.72 |
| SE(3) Trans w/MLM | 67.65 ± 1.31 | 0.55 ± 0.03 | 77.54 ± 0.06 | 87.08 ± 1.79 | 0.681 ± 0.109 | 43.20 ± 4.63 |
| SE(3) Trans w/MVCL | 44.01 ± 0.72 | 0.20 ± 0.00 | 75.59 ± 2.54 | 83.70 ± 0.04 | 1.110 ± 0.074 | 34.34 ± 4.76 |
| SE(3) Trans w/PFP | 69.27 ± 2.55 | 0.81 ± 0.17 | 77.60 ± 0.39 | 87.06 ± 0.66 | 0.536 ± 0.047 | 47.94 ± 3.51 |
| ProtBERT w/MLM | 95.45 ± 0.10 | 9.37 ± 0.62 | 77.09 ± 0.36 | 85.83 ± 0.73 | 0.511 ± 0.011 | 49.81 ± 1.32 |
| ProtBERT w/MVCL | 95.78 ± 0.21 | 28.51 ± 0.06 | 73.58 ± 0.13 | 83.40 ± 0.33 | 0.511 ± 0.008 | 50.25 ± 1.76 |
| ProtBERT w/PFP | 64.25 ± 0.35 | 0.36 ± 0.00 | 78.79 ± 0.85 | 87.41 ± 1.00 | 0.529 ± 0.018 | 49.99 ± 1.77 |
| GVP w/MLM | 89.09 ± 0.00 | 11.76 ± 0.01 | 72.54 ± 0.06 | 79.03 ± 0.01 | 0.438 ± 0.013 | 47.28 ± 0.00 |
| GVP w/MVCL | 88.68 ± 0.00 | 10.96 ± 0.01 | 71.45 ± 0.42 | 78.19 ± 0.13 | 0.423 ± 0.011 | 59.31 ± 0.00 |
| GVP w/PFP | 71.83 ± 0.01 | 1.50 ± 0.00 | 72.72 ± 0.36 | 79.94 ± 0.58 | 0.431 ± 0.002 | 59.04 ± 0.00 |
| D-Trans w/MLM | 94.40 ± 1.13 | 29.93 ± 0.09 | 74.46 ± 0.78 | 81.44 ± 0.59 | 0.464 ± 0.023 | 55.79 ± 0.50 |
| D-Trans w/MVCL | 98.11 ± 0.28 | 36.50 ± 3.17 | 73.26 ± 0.18 | 81.25 ± 0.50 | 0.437 ± 0.002 | 59.91 ± 0.13 |
| D-Trans w/PFP | 96.65 ± 0.21 | 34.55 ± 2.53 | 73.50 ± 0.30 | 82.64 ± 0.05 | 0.441 ± 0.020 | 59.66 ± 0.75 |
| ESM-2 | 93.10 ± 1.71 | 55.09 ± 0.22 | 81.25 ± 1.44 | 88.17 ± 1.10 | 0.416 ± 0.011 | 59.59 ± 0.35 |
| ESM-C | 94.42 ± 0.10 | 35.25 ± 2.48 | 79.81 ± 0.36 | 86.65 ± 0.37 | 0.461 ± 0.00 | 53.35 ± 0.84 |
| SaProt | 94.75 ± 0.00 | 57.35 ± 0.01 | 81.43 ± 0.90 | 87.16 ± 0.13 | 0.423 ± 0.008 | 58.81 ± 0.02 |

To provide a more comprehensive evaluation, we perform full parameter fine-tuning of the pretrained models on three downstream tasks, namely PCS, PROTACs, and PLI, with all encoder parameters updated during training. The corresponding results are presented in the Tab. 6.

> **Observation 1**. Full-parameter fine-tuning has mixed effects across different models, but overall leads to slightly lower performance compared to the encoder-freezing setting.

Table 7: Results on KIBA Dataset.

| | EGNN | EGNN w/PFP | SE(3) Trans | SE(3) Trans w/MVCL | ProtBERT | ProtBERT w/MVCL | ESM-2 |
|---|---|---|---|---|---|---|---|
| MSE↓ | 0.434 ± 0.007 | 0.396 ± 0.006 | 0.442 ± 0.034 | 0.473 ± 0.026 | 0.471 ± 0.005 | 0.393 ± 0.001 | 0.454 ± 0.004 |
| Pear(%)↑ | 65.09 ± 0.68 | 65.27 ± 0.15 | 63.19 ± 1.15 | 58.57 ± 1.84 | 60.89 ± 3.66 | 65.10 ± 0.76 | 58.14 ± 0.01 |

We additionally report the performance of EGNN, SE(3)-Transformer, ProteinBERT, and ESM-2 on the PLI task using the KIBA dataset, as shown in Tab. 7. Compared with the result on the DAVIS dataset, we observe that as the dataset size increases from 14,464 to 89,958, the performance of all models on the PLI task improves to varying degrees. Notably, the SE(3) Transformer benefits little from pretraining, whereas ProteinBERT shows a moderate performance gain due to pretraining.

> **Observation 2**. Model performance on downstream tasks like PLI consistently improves with increased data size, indicating a power-law-like scaling trend.

Furthermore, we visualize the relationship between model size and performance across four downstream tasks, as shown in Fig. 2 and Fig. 3. Our observation is summarized as follows:

**Observation 3**. Large-scale models such as SaProt, ESM-2 and ESM-C demonstrate clear advantages on tasks that require only a single protein sequence as input, such as PFA and MTP. However, when downstream tasks involve more complex inputs, e.g., the PROTACs task, which implicitly requires the alignment between protein and compound representations, smaller models like D-Transformer and EGNN tend to perform better, suggesting their stronger capacity to handle multi-modal integration despite having fewer parameters.

**Observation 4**. For domain-specific models, we do not observe clear evidence that larger model size consistently leads to better performance across downstream tasks. In the PROTACs task, both domain-specific models perform poorly. However, in the PLI and GO tasks, these models achieve strong results despite their relatively small sizes, suggesting that domain knowledge may play a more critical role than scale in certain contexts.

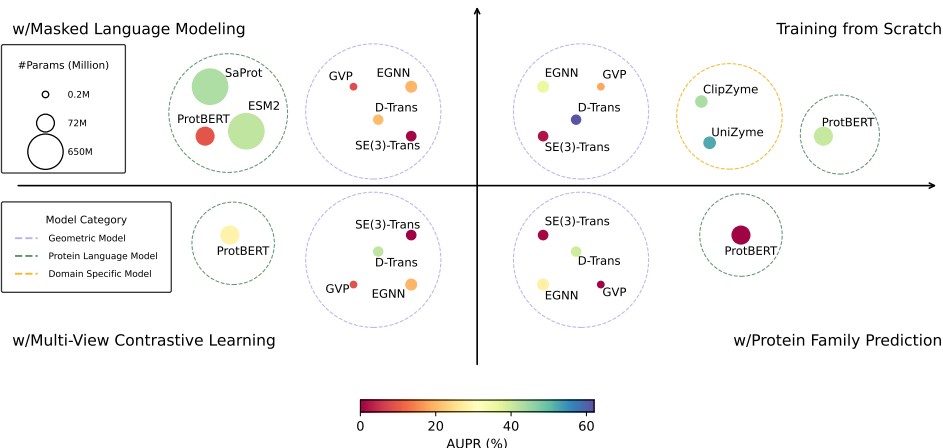

(a) Comparisons between general and domain models on protein cleavage site prediction (PCS).

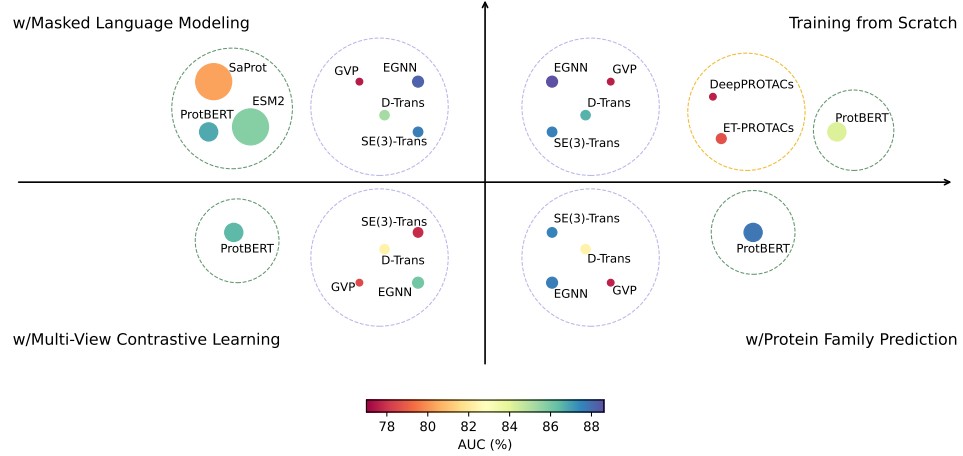

(b) Comparisons between general and domain models on targeted protein degradation by proteolysis-targeting chimeras (PROTACs).

Figure 2: Visualization of model size and performance across two specialized tasks: (a) PCS and (b) PROTACs. Each point represents a model, with size indicating parameter count and color intensity reflecting task performance. Each point represents a model, with its size corresponding to the number of parameters, ranging from 2M for GVP to 650M for ESM. Larger points indicate larger models. The color intensity of each point indicates performance, with darker shades representing stronger results. The dashed circular boundaries group models by category: models incorporating geometric information in color, protein language models in color, and domain-specific models in color.

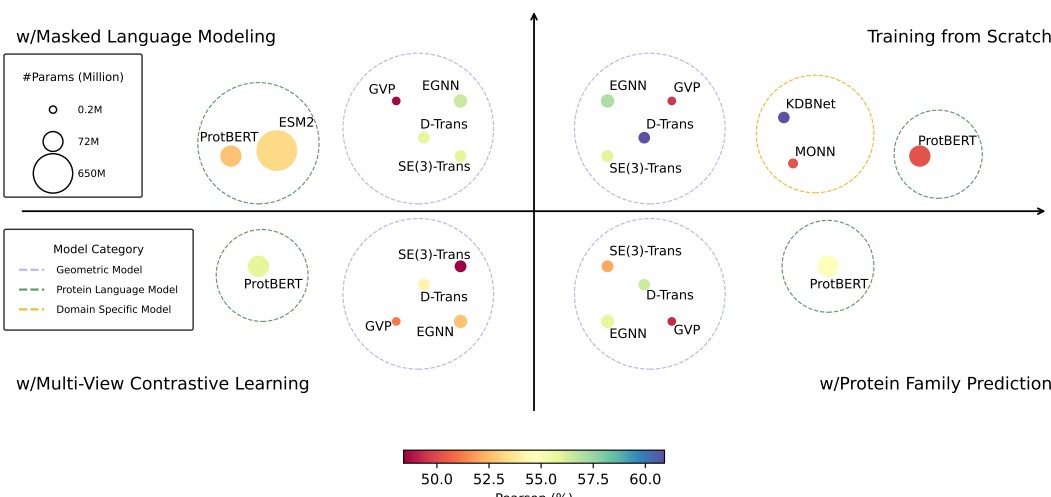

(a) Comparisons between general and domain models on Protein-Ligand Interaction (PLI).

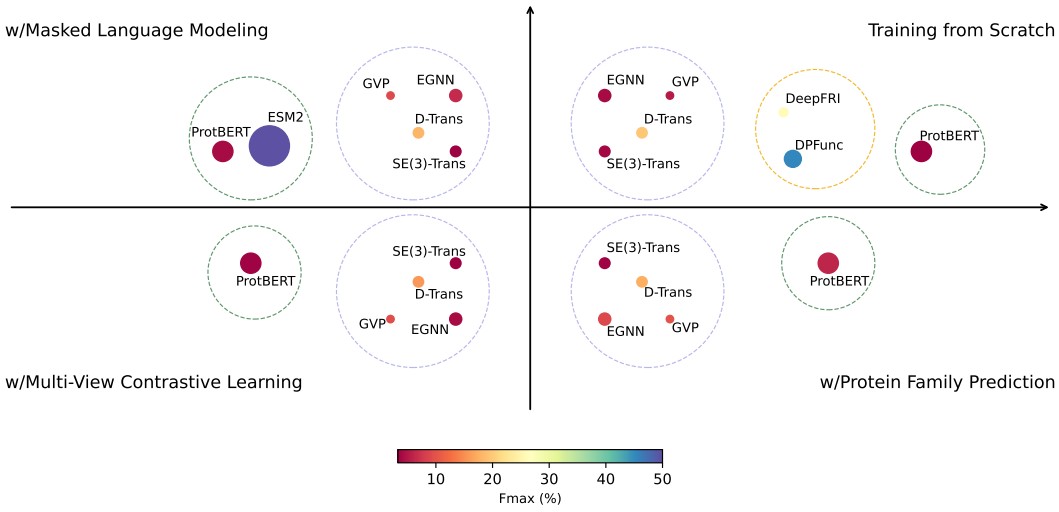

(b) Comparisons between general and domain models on protein function prediction (PFA).

Figure 3: Visualization of model size and performance across two general tasks: (a) PLI and (b) PFA. Each point represents a model, with size indicating parameter count and color intensity reflecting task performance.

## B  RELATED WORKS

Existing benchmarks for protein modeling primarily emphasize sequence-based models. TAPE (Rao et al., 2019) introduces standardized sequence-centric tasks, including structure prediction and remote homology detection. PEER (Xu et al., 2022) expands the sequence-based scope to cover protein–protein interactions and functional annotations. ProteinGLUE (Capel et al., 2022) is another sequence-based benchmark designed to evaluate sequence pretraining methods. ProteinGym (Notin et al., 2023) establishes an evaluation framework for mutation effect prediction with protein language models. ProteinBench (YE et al., 2025) evaluates protein foundation models mainly in protein generation tasks. Among existing benchmarks, ProteinWorkshop (Jamasb et al., 2024) incorporates the equivariant graph neural networks to evaluate the structure-based models. However, ProteinWorkshop lacks a unified benchmarking pipeline that systematically compares sequence-based, structure-based, and hybrid models within a standardized framework.

Furthermore, existing benchmarks predominantly focus on general model architectures such as LSTMs (Hochreiter & Schmidhuber, 1997), Transformers (Vaswani et al., 2017), and EG-NNs (Satorras et al., 2021). In contrast, Protap explores the advantages of domain-specific model designs and the integration of domain knowledge. Our Protap systematically benchmarks domain-specific models alongside pretrained models with general architectures. This can provide deeper insights into effective model design and performance. Moreover, Protap introduces two novel specialized applications: enzyme-catalyzed protein cleavage site prediction and targeted protein degradation by PROTACs, which are biological processes not considered by prior benchmarks. A more comprehensive comparison between existing benchmarks is presented in Tab. 1.

## C  PRELIMINARIES

### C.1  PROTEIN DATA

Proteins obtained through high-throughput sequencing are typically represented as sequences. However, using laboratory techniques such as X-ray crystallography, electron microscopy, XFEL, and nuclear magnetic resonance, the 3D structures of proteins can be elucidated. Unlike Computer Vision and Natural Language Processing, proteins can be characterized using various descriptions, such as 1D sequences, 2D graphs, and 3D structures.

**Sequence**. The most common representation is the amino acid sequence, typically in FASTA format using single-letter codes. Sequence-based models, originally RNNs and CNNs, are now dominated by Transformers due to their superior long-range modeling. Large pre-trained models such as ESM-2 (Rives et al., 2021; Lin et al., 2023), ProteinBERT (Brandes et al., 2022), and ProtTrans (Elnaggar et al., 2021) capture biochemical properties and evolutionary relationships, with structural information implicitly encoded in their learned representations.

**2D Graph.**. Proteins can also be described as molecular graphs, with residues as nodes and chemical or spatial relationships as edges. 2D graphs capture connectivity and topological features but lack explicit spatial details. Contact maps and models like GraphFormer (Ying et al., 2021) encode residue-residue proximity or graph connectivity into the attention mechanism, partially reflecting structural information.

**3D Structure**. Many methods have been developed for modeling the 3D structures of proteins. 3D CNNs employ convolutional neural networks to extract local spatial features of proteins. Graph neural networks are widely used for representing protein and molecular structures, with examples including proteinMPNN (Dauparas et al., 2022), SE(3)-Transformer (Fuchs et al., 2020), GVP (Jing et al., 2021), and GearNet (Zhang et al., 2023). GearNet applies contrastive learning for the pre-training of protein structures. Additionally, Transformer-M (Luo et al., 2023a) incorporates 3D information through distance matrices, enabling transformers to represent protein 3D structures while maintaining invariance. More recently, some works have begun to include protein surface information in modeling; for instance, ProteinINR (Lee et al., 2024) integrates sequence, 3D structure, and surface information to enhance protein representations.

## C.2 TERMINOLOGY

This section provides detailed definitions of the professional terms used in the paper.

**Protein Family**. A protein family is a group of evolutionarily related proteins that typically share similar amino acid sequences, three-dimensional structures, and biological functions. Most members of a protein family are encoded by genes from a corresponding gene family, where each gene-protein pair has a one-to-one relationship. To date, more than 60,000 protein families have been identified, though exact numbers vary depending on the criteria and classification methods used.

**Enzyme-catalyzed Protein Cleavage**. Proteolytic enzymes first recognize short sequences or structural motifs within a substrate protein and then hydrolyze the peptide bond at the corresponding cleavage site, thereby fragmenting the polypeptide chain. This tightly regulated process governs protein maturation, turnover, and signaling, and its accurate in-silico prediction assists enzyme engineering and therapeutic target identification.

**Proteolysis-Targeting Chimeras (PROTACs)**. A PROTAC is a heterobifunctional small molecule comprising (i) a warhead that binds the target (disease-relevant) protein, (ii) a chemical linker, and (iii) an E3-ligase-recruiting ligand. By simultaneously engaging the target protein and an E3 ubiquitin ligase, a PROTAC forms a ternary complex that drives poly-ubiquitination of the target, leading to its proteasomal degradation. Because PROTACs act catalytically and do not rely on classical active-site inhibition, they can eliminate previously "undruggable" proteins and often function at lower doses than conventional inhibitors.

**Protein-Ligand Binding Affinity**. Protein–ligand binding denotes the selective, high-affinity association between a protein and a small-molecule ligand to form a stable complex. Binding strength is quantified by the change in Gibbs free energy ($\Delta$G) or related measures such as the dissociation constant ($K_d$); a more negative $\Delta$G (or lower $K_d$) reflects stronger affinity. Reliable computational estimation of binding affinity accelerates virtual screening and lead optimization in drug discovery by prioritizing ligands most likely to bind their target proteins.

**Gene Ontology (GO)**. Gene Ontology is a standardized vocabulary used to describe the functions of genes and proteins in a consistent and structured way. It provides a common language that allows researchers across different species and databases to describe what a gene product does, where it does it, and what biological processes it is involved in. GO is organized into three main categories, known as ontologies: (i) Biological Process (BP), which describes the broader biological goals that a gene or protein contributes to. (ii) Molecular Function (MF), which defines the specific biochemical activity of the gene product. (iii) Cellular Component (CC), which indicates the location within the cell where the gene product is active. Each GO term is assigned a unique identifier, e.g., GO: 0003677 for "DNA binding".

# D DETAIL OF DOWNSTREAM APPLICATION AND DATASET

## D.1 ENZYME-CATALYZED PROTEIN CLEAVAGE SITE PREDICTION

**Task Definition**. Cleavage site prediction is formulated as a large-scale, imbalanced, residue-level binary classification problem. Let $P_s$ denote a substrate protein of length $|R|$, and for each residue $r_j$ ($j = 1, \ldots, |R|$), let $x_j \in \mathbb{R}^d$ be its feature vector. The objective is to learn a function

$$f : \mathbb{R}^d \longrightarrow \{0, 1\}$$

that outputs 1 if $r_j$ is a cleavage site under catalysis by a given enzyme, and 0 otherwise. Formally, each substrate $P_s$ is associated with a label vector $Y \in \{0, 1\}^{|R|}$, where $Y_j = 1$ indicates cleavage at residue $j$. Training proceeds over residue-level examples $\{(x_j, Y_j)\}$ sampled from annotated cleavage datasets, enabling model generalization to varied substrate contexts.

**Settings**. For all pretrained models, we retain the same architectural hyperparameters as used during pretraining. During downstream training, we use a unified configuration across all models: the number of training epochs is set to 50, the learning rate to $1 \times 10^{-4}$, and the batch size to 24. A cosine annealing scheduler is employed to adjust the learning rate over time. To ensure robustness and reproducibility, we fix a set of five random seeds $\{42, 128, 256, 512, 1024\}$. For each model, we report the mean and standard deviation of its performance across these five runs. All experiments

were conducted using 8 NVIDIA L40 GPUs. The average training time varied depending on the model size and task complexity, ranging from approximately 15 minutes to 5 hours.

**Metric**. The evaluation of enzyme-catalyzed protein cleavage site prediction is framed as a residue-level binary classification task under substantial class imbalance. Accordingly, the area under the receiver operating characteristic curve (AUC) and the area under the precision-recall curve (AUPR) are used as primary evaluation metrics. AUC quantifies the trade-off between true positive rate (TPR) and false positive rate (FPR) across various classification thresholds, offering a global view of discriminative ability. However, due to the rarity of cleavage sites, AUPR is considered more informative, as it emphasizes the precision-recall behavior specific to the minority class. In particular, AUPR reflects the expected precision over all recall levels, which is critical for the reliable detection of enzymatic cleavage sites.

**Dataset**. The detailed information of the dataset is as follows:

- **Data Source**. The original data are from the MEROPS (Rawlings et al., 2014) database. This dataset has a public website, available at the following address: `http://cadd.zju.edu.cn/protacdb/`

- **Pre-process**.

  - Prior studies (Rawlings et al., 2014) have shown that slight sequence variations among enzymes within the same MEROPS category are negligible. As a result, we generalize the hydrolysis data from a specific substrate-enzyme interaction to all enzymes in that category. This allowed us to augment our dataset by linking each substrate not only to its originally annotated enzyme but also to other enzymes classified in the same MEROPS group. Finally, we selected two enzyme families, `C14.003` and `M10.003`, for evaluation.

  - We performed quality control on the raw data by filtering out entries with hydrolysis sites exceeding the maximum sequence length.

  - We converted the structural data of proteins in the dataset into a dictionary format and stored it in a `.pickle` file. The fields are as follows:

    ```
    {
    "Q5QJ38":{
        "name": "Q5QJ38",
        "seq": "MPQLLRNVLCVIETFHKYASEDSNGAT...",
        "coords": [[[-0.432, 25.507, -8.242], ...], ...],
        "cleave_site": [136]
        }
    }
    ```

- **Data Format**. The sequence and structure of substrate proteins are stored in a `.pickle` file. The hydrolysis site information of substrate proteins is stored in a `.pickle` file as follows. It is worth noting that the hydrolysis site positions are 1-based indexed.

    ```
    {
    "P31001_MER0000622": [110, 263],
    "O00232_MER0000622": [276, 334, 19],
    ...
    }
    ```

- **Data Statistics**. The statistics of the dataset after preprocessing are summarized in Table 8. The training and test sets were split based on sequence similarity, ensuring that the sequence similarity between the test and training sets is below 60%.

Table 8: Statistics of the dataset after preprocessing.

| Enzyme Family | #Train | #Test | Cleavage Sites (Train) | Cleavage Sites (Test) |
|---|---|---|---|---|
| C14.005 | 468 | 117 | 656 | 159 |
| M10.003 | 375 | 92 | 953 | 157 |

- **Usage**. This dataset is used for a residue-level binary classification task. The goal is to predict the hydrolysis sites of a protein by a given enzyme family, based on its sequence or structure.

- **License**. We release a preprocessed version of the dataset under the MIT License. The original MEROPS database is provided under the terms of the GNU Library General Public License and is available at: `https://www.ebi.ac.uk/merops/about/availability.shtml`.

## D.2 Targeted Protein Degradation by Proteolysis-Targeting Chimeras

**Task Definition**. This task is typically formulated as a binary classification problem. Given a PRO-TAC candidate and its corresponding POI and recruited E3 ligase, the goal is to predict whether the induced ternary complex will successfully trigger degradation. Formally, let the PROTAC molecule be decomposed into three modular components: warhead $w$, linker $l$, and E3-ligand $el$. Along with the protein of interest $p$ and the E3 ligase $eg$, each component is processed through dedicated neural encoders:

$$\mathbf{h}_p = f_p(p), \quad \mathbf{h}_w = f_w(w), \quad \mathbf{h}_l = f_l(l), \quad \mathbf{h}_{el} = f_{el}(el), \quad \mathbf{h}_{eg} = f_{eg}(eg)$$

The resulting latent representations are concatenated to form a joint embedding that captures the structural and biochemical context of the ternary complex:

$$\mathbf{h}_{\text{concat}} = \text{concat}(\mathbf{h}_p, \mathbf{h}_w, \mathbf{h}_l, \mathbf{h}_{el}, \mathbf{h}_{eg})$$

This composite feature vector is passed through a multi-layer perceptron (MLP) with two fully connected layers to output the degradation prediction score:

$$\hat{y} = \sigma(\text{MLP}(\mathbf{h}_{\text{concat}})) \in [0, 1]$$

where $\hat{y}$ denotes the predicted probability of successful POI degradation. The model is trained using binary cross-entropy loss, with ground truth labels indicating degradation activity.

**Settings**. For this task, we retain the same architectural hyperparameter configuration as used during pretraining. To encode SMILES representations, we adopt the GVP (Jing et al., 2021) architecture with a node dimensionality of 128, an edge dimensionality of 32, and a total of 3 layers. Models are trained for 50 epochs with a learning rate of 5e-4, a batch size of 24, and the Adam optimizer. We report the mean and standard deviation of performance across a fixed set of random seeds. All experiments are conducted on NVIDIA L40 GPUs, with per-run training time ranging from approximately 20 minutes to 6 hours, depending on the model and task complexity.

**Metrics**. The evaluation of PROTAC-induced protein degradation prediction is framed as a binary classification task. Accordingly, we adopt two widely used classification metrics: Accuracy measures the proportion of correct predictions among all samples, providing a straightforward indication of overall model performance. Area Under the Receiver Operating Characteristic Curve (AUC) quantifies the model's ability to distinguish degraders from non-degraders across varying decision thresholds.

**Dataset**. The detailed information of the dataset is as follows:

- **Data Source**. The original data was collected from the PROTAC-DB (Ge et al., 2025). This dataset has a public website, available at the following address: `http://cadd.zju.edu.cn/protacdb/`

- **Pre-process**.
  - Following the processing strategy proposed by ET-PROTACs (Cai et al., 2025), we assign a degradation label to each triplet in the dataset. Specifically, we first examine whether a PROTAC entry contains $DC_{50}$ measurements. If available, entries with $DC_{50}$ values below 1000 nM are labeled as active, while those equal to or above 1000 nM are labeled as inactive. For entries lacking $DC_{50}$ data, we instead inspect the reported degradation percentage of the target protein. If available, samples with degradation rates of at least 70% are considered active, and those below 70% are labeled as inactive. In cases where neither $DC_{50}$ nor degradation percentage is reported, we fall back to $IC_{50}$ values. Similarly, entries with $IC_{50}$ below 1000 nM are labeled as active, and those above or equal to this threshold are labeled as inactive.
  - For the target protein and E3 ligase, we retained only the entries with available structures in the AFDB. We filtered out the target proteins with the following UniProt IDs: P36969, P03436,

P0DTD1, Q12830, and G0R7E2. The structural data of the target proteins and E3 ligases are obtained from AFDB. Following the approach in (Ingraham et al., 2019), we further convert the format into a dictionary and store it in a .json file. The fields are as follows:

```
[
"Q8IWV7":{
        "seq": "MADEEAGGTERMEISAELPQTPQRLASWWDQQVDFYTA...",
        "coord": [[[21.9960, 68.3170,-49.9029], ...],...]
    }
    ...
]
```

For each structure, coord is a nested list of shape $(N \times 4 \times 3)$, representing the 3D coordinates of the backbone atoms N, C$\alpha$, C, and O for each residue in order, and $N$ is the length of the amino acid sequence.

– For the warhead, linker, and E3-ligand, we use their SMILES strings to generate 20 conformers with RDKit's ETKDGv3 method. The conformers are optimized using a force field, and the lowest-energy conformer is selected. The final structure is saved in an SDF file.

- **Data Format**. The protein data, including the name, sequence, and coordinates, is stored in .json format. The structures of the warhead, linker, and E3-ligand are stored in SDF files. The label data is saved in a .txt file with the following format. The content below shows only key fields; for the complete list of fields, please refer to the full file.

```
uniprot  e3_ligase_structure linker_sdf    warhead_sdf   e3_ligand_sdf   label
Q00987      Q96SW2            linker_2.sdf  warhead_7.sdf  e3_ligand_7.sdf    1
Q00987      Q96SW2            linker_2.sdf  warhead_7.sdf  e3_ligand_16.sdf   1
P10275      P40337            linker_13.sdf warhead_27.sdf e3_ligand_27.sdf   0
P10275      P40337            linker_33.sdf warhead_27.sdf e3_ligand_27.sdf   0
```

- **Data Statistics**. The number of each component in the dataset is summarized in Table 9. The dataset was randomly split into a training set and a test set, with 80% used for training and 20% for testing.

Table 9: Statistics of PROTACs dataset components.

| Object | Target Protein | E3-Ligase | Warhead | Linker | E3-Ligand | Label: 1 | Label: 0 |
|--------|---------------|-----------|---------|--------|-----------|----------|----------|
| **Number** | 154 | 10 | 552 | 1,534 | 131 | 2,244 | 1,878 |

- **Usage**. Based on the preprocessed reaction labels, this dataset is used for a binary classification task, where the goal is to predict the targeted degradation effect of PROTACs given the information of the target protein, E3 ligase, and the PROTACs.

- **License**. We release a preprocessed version of the dataset under the MIT License. Please refer to the usage license of the original data at: http://cadd.zju.edu.cn/protacdb/downloads, and follow the original authors' terms of use.

## D.3  PROTEIN–LIGAND INTERACTIONS

**Task Definition**. Protein-ligand binding is the process by which proteins or various small molecules interact with high specificity and affinity to form a particular complex. A thorough understanding of the mechanisms underlying protein-ligand interactions is a prerequisite for gaining in-depth insights into protein function. The goal of rational drug design is to leverage structural data and knowledge of protein-ligand binding mechanisms to optimize the process of discovering new drugs. The driving force for protein-ligand binding arises from a combination of interactions and energy exchanges among proteins, ligands, water molecules, and buffering ions. Because the extent of protein-ligand binding is determined by the magnitude of negative $\Delta G$, $\Delta G$ can be considered to determine the stability of any given protein-ligand complex, or equivalently, the binding affinity of a ligand for a given receptor. Experimentally measuring protein-ligand binding affinity is both time-consuming and complex, making it impractical to rely solely on experimental approaches for drug discovery from large compound libraries. In computational medicinal chemistry, predicting ligand binding affinity remains an open challenge. Existing deep learning methods attempt to directly predict binding affinity using affinity data from databases such as PDBBind.

From a machine learning perspective, protein-ligand affinity prediction is formulated as a regression problem, where the goal is to learn a function

$$f : \mathbb{R}^d \to \mathbb{R}$$

that maps the given input features $X_i$ (e.g., protein structural or sequence data, ligand chemical descriptors, etc.) to a continuous value $y_i \in \mathbb{R}$, representing the quantitative binding affinity. Concretely, each protein-ligand pair $(P_i, l_i)$ is associated with a feature vector $X_i \in \mathbb{R}^d$ and an affinity value $y_i$. The learning objective is to minimize the discrepancy between the predicted affinity $\hat{y}_i = f(X_i)$ and the experimentally measured (or otherwise ground-truth) value $y_i$, typically via metrics such as root mean squared error (RMSE) or mean absolute error (MAE). The function $f$ can be learned using a training dataset

$$\{(X_i, y_i)\}_{i=1}^n,$$

and subsequently evaluated on unseen data to gauge its predictive performance and generalizability, ultimately utilized for drug screening.

**Settings**. For this task, we retain the same architectural hyperparameter configuration as used during pretraining. To encode SMILES representations, we adopt the GVP (Jing et al., 2021) architecture with a node dimensionality of 128, an edge dimensionality of 32, and a total of 3 layers. The maximum length of the pocket amino acid sequence is set to 85 residues, with shorter sequences padded and longer sequences truncated accordingly. Models are trained for 50 epochs with a learning rate of 5e-4, a batch size of 48 or 96, and the Adam optimizer. We report the mean and standard deviation of performance across a fixed set of random seeds. All experiments are conducted on NVIDIA L40 GPUs, with per-run training times ranging from approximately 20 minutes to 5 hours, depending on the model and task complexity.

**Metrics**. To assess a model's ability to predict binding affinity accurately, we adopt two complementary metrics: Mean Squared Error (MSE) and Pearson correlation coefficient. A lower MSE indicates more accurate regression of affinity values, while a higher Pearson coefficient reflects stronger linear correlation between predicted and true affinities. These metrics jointly capture both the precision and consistency of model predictions, providing a comprehensive evaluation of regression performance.

**Dataset**. The detailed information of the dataset is as follows:

- **Original Data Source.** The original data was collected from the KDBNet (Luo et al., 2023b). The data is stored at `https://www.dropbox.com/s/owc45bzbfn05ix4/data.tar.gz`. Based on the existing DAVIS (Davis et al., 2011) and KIBA (Tang et al., 2014) datasets, the authors further extracted the binding pockets of proteins from protein-ligand complexes for use in modeling this task.

- **Preprocess.**

  - In the DAVIS portion of the KDBNet dataset, the protein `6FDY.U` contains missing coordinate values, which we filter out.

  - Following the approach in (Ingraham et al., 2019), we convert the format of the protein structural data provided by the authors and store it as a dictionary in a JSON file:

    ```
    [
    "4WSQ.B":{
            "uniprot_id": "Q2M2I8",
            "seq": "EVLAEGGFAIVFLCALKRMVCKREIQIMRDLS...",
            "coord": [[[6.6065,16.2524,52.3289], ...], ...]
        }
        ...
    ]
    ```

    For each structure, `coords` is a nested list of shape *(N × 4 × 3)*, representing the 3D coordinates of the backbone atoms N, C$\alpha$, C, and O for each residue in order, and $N$ is the length of the amino acid sequence.

- **Format.** The protein data, including the name, UniProt ID, sequence, and coordinates, is stored in `.json` format. The label data is saved in a `.txt` file with the following format:

```
                drug       protein      Kd         y          protein_pdb
        0       5291       AAK1         10000.0    5.0        4WSQ.B
        1       5291       ABL1p        10000.0    5.0        3QRJ.B
        2       5291       ABL2         10.0       7.99568    2XYN.C
```
The ligand data is stored in SDF format.

- **Statistics.** As shown in Table 10, DAVIS contains 226 proteins, 64 compounds, and 14,464 interaction pairs, while KIBA includes 160 proteins, 1,986 compounds, and 89,958 pairs. The dataset was randomly split into a training set and a test set, with 80% used for training and 20% for testing. These datasets vary in scale and compound diversity, providing a comprehensive benchmark for model evaluation.

Table 10: Statistics of protein-ligand datasets.

| Dataset | Protein | Ligand | Pair |
|---------|---------|--------|--------|
| DAVIS   | 226     | 64     | 14,464 |
| KIBA    | 160     | 1,986  | 89,958 |

- **Usage.** This dataset is used for a regression task, where the goal is to predict the binding affinity for each protein-ligand pair.
- **License.** We release a preprocessed version of the dataset under the MIT License. The original dataset, also licensed under the MIT License, is available at: `https://github.com/luoyunan/KDBNet/blob/main/LICENSE`.

### D.4 PROTEIN FUNCTION ANNOTATION PREDICTION

**Task Definition**. From a computational standpoint, protein function prediction is inherently a large-scale, sparse, and imbalanced multi-label classification problem. The goal is to predict multiple output labels from the given input features $X_i$. Let the set of labels be

$$\mathcal{L} = \{BP, MF, CC\}$$

Thus, each protein $P_i$ is associated with a label vector $Y_i \in \{0, 1\}^{|\mathcal{L}|}$, where $Y_i[j] = 1$ indicates that protein $P_i$ is annotated with the $j$-th label, and $Y_i[j] = 0$ indicates it is not.

The goal is to learn a mapping function $f(X_i)$ to predict the label vector $Y_i$, i.e.:

$$f : \mathbb{R}^d \to \{0, 1\}^{|\mathcal{L}|}$$

This function can be learned using the training dataset $\{(X_i, Y_i)\}_{i=1}^n$.

**Settings**. For this task, to rigorously assess the model's generalization ability, we included only those test sequences whose similarity to the training set was below 50%. we adopt the same architectural hyperparameter settings as used during pretraining to ensure consistency. Each model is trained for 50 epochs using the Adam optimizer with a learning rate of 5e-4 and a batch size of 24. To ensure robustness, we report the average performance and standard deviation across a predefined set of random seeds. All training is conducted on NVIDIA L40 GPUs, with individual runs taking between 2 minutes and 10 hours, depending on the model architecture and task complexity.

**Metrics**. Given the inherent sparsity of function annotation labels, where each protein is typically associated with only a small subset of possible functions, we evaluate model performance using Fmax and AUPR (Area Under the Precision-Recall Curve). These metrics are particularly suited for imbalanced multilabel classification tasks, where a higher Fmax and AUPR indicate better predictive capability in accurately identifying relevant functional annotations.

**Dataset**. The detailed information of the dataset is as follows:

- **Data Source**. The original data are from DeepFRI (Gligorijević et al., 2021)(`https://github.com/flatironinstitute/DeepFRI`), and the corresponding structural data are collected by ProteinWorkshop (Jamasb et al., 2024)(`https://zenodo.org/records/8282470/files/GeneOntology.tar.gz?download=1`).
- **Pre-process**.

- We performed quality control on the raw data by filtering out entries with missing coordinates or with all function labels equal to 0.
- Based on the label annotations and protein information provided in the original dataset, we unified each protein entry into a dictionary format containing its name, sequence, coordinates, and functional labels. The final data was saved in a `.json` file, with each entry structured as follows:

```
[
  {
    "name": "2P1Z-A",
    "seq": "SKKAELAELVKELAVYVDLRRATLHARASRLIGELLRELTADWDYVA...",
    "coords": [[[6.4359, 51.3870, 15.4490], ...], ...],
    "molecular_function": [0, 0, ...1,...],
    "biological_process": [0, 0, ...0,...],
    "cellular_component": [0, 0, ...1,...]
  },
  ...
]
```

The fields `molecular_function`, `biological_process`, and `cellular_component` store one-hot encoded functional annotations. If you need Gene Ontology term IDs corresponding to the functional labels, please refer to the `label.tsv` file. The order of entries in this file strictly matches the position of each label in the one-hot vectors.

- **Data Format**. The protein name, sequence, coordinates, and functional labels are stored in a `.json` file. The corresponding Gene Ontology terms for the functional labels are provided in the `label.tsv` file.

- **Data Statistics**. Since *molecular function* defines the biological roles that a protein participates in, we restrict our functional prediction evaluation to *molecular function* only. The following Table 11 summarizes the label distribution across all proteins in the dataset. Label 1 refers to the total number of positive functional labels associated with proteins. Label 0 refers to the total number of functional labels not associated with the proteins.

Table 11: Statistics of positive and negative functional labels in the training and test sets.

| Objective | #Train | #Test | Label: 1 (Train) | Label: 0 (Train) | Label: 1 (Test) | Label: 0 (Test) |
|---|---|---|---|---|---|---|
| **Number** | 23760 | 202 | 121,650 | 11,496,990 | 17,709 | 971,538 |

- **Usage**. This dataset is used for a multi-label classification task, where the goal is to predict the functional labels of each protein based on its sequence and structural information.

- **License**. We release a preprocessed version of the dataset under the MIT License. The original dataset is available under a BSD 3 license at https://github.com/flatironinstitute/DeepFRI/blob/master/LICENSE. And the protein structure data is released under the CC BY 4.0 license and is available at: https://openreview.net/pdf?id=sTYuRVrdK3

### D.5 MUTATION EFFECT PREDICTION FOR PROTEIN OPTIMIZATION

**Task definition**. The log-ratio between wild-type and mutant amino acid probabilities has been shown to be an effective estimator of mutational impact (Riesselman et al., 2018; Notin et al., 2022). In the zero-shot setting, we do not access any label information. Instead, we perform inference using a model pretrained with masked language modeling (MLM). The goal is to quantify the log-likelihood of protein variants under the background. The calculation is shown in the equation.

$$\sum_{t \in T} \log p(x_t = x_t^{mt}|S_{-t}) - \log p(x_t = x_t^{wt}|S_{-t})$$

where $T$ is a set of positions where multiple mutations exist in the same sequence, and $S$ is the wild-type sequence.

**Settings**. In the zero-shot setting, we do not use any label information, nor do we perform further training or fine-tuning. The prediction of mutation effects relies on the model's estimated probabilities for the mutant and wild-type amino acids at the mutation site. Therefore, pretraining methods whose models do not expose logits are unsuitable for this zero-shot setting, and no additional domain-specific models are incorporated in this study. Other types of pretraining methods are not suitable for this zero-shot scenario, and no additional domain-specific models are introduced in this application.

**Metrics**. Due to the non-linear relationship between protein function and fitness, the Pearson correlation coefficient is a suitable metric for evaluating model performance. Another evaluation metric is AUC, which assesses the model's ability to rank and discriminate between functionally neutral and deleterious mutations. During AUC calculation, `DMS_Score_Bin` is used as the ground-truth label (1 = positive, 0 = negative), while the model's continuous prediction scores serve as the decision function.

**Dataset**. The detailed information of the dataset is as follows:

• **Data Source**. The dataset we used to evaluate in this benchmark is from ProteinGym (Gligorijević et al., 2021)(`https://proteingym.org/`)

• **Pre-process**. Structural data is predicted by OmegaFold (Wu et al., 2022), where the input sequences for structure prediction are the wild-type sequences. We removed samples for which OmegaFold failed to generate structural predictions due to excessive sequence length.

• **Data Format**. The DMS substitution data provided by ProteinGym is formatted as shown below. The first column contains the mutation information, using 1-based indexing. For example, `F1I` indicates that the first amino acid in the wild-type sequence, originally phenylalanine (F), is mutated to isoleucine (I).

```
mutant      mutated_sequence                DMS_score DMS_score_bin
 F1I     ITLIELMIVIAIVGILAAVALPAYQDYTA...    -3.598          0
 F1L     LTLIELMIVIAIVGILAAVALPAYQDYTA...    -0.678          0
 F1V     VTLIELMIVIAIVGILAAVALPAYQDYTA...     1.299          1
 F1S     STLIELMIVIAIVGILAAVALPAYQDYTA...    -0.127          0
 T2A     FALIELMIVIAIVGILAAVALPAYQDYTA...     0.786          1
```

• **Data Statistics**. After preprocessing, the dataset contains 201 proteins and 2,413,913 mutations. The detailed statistics of the preprocessed dataset are shown in Table 12.

Table 12: Overall statistics of the ProteinGym DMS substitution dataset.

| Objective | Proteins | Mutants |
|---|---|---|
| **Number** | 201 | 2,413,913 |

• **Hosting**. The ProteinGym dataset is well-organized, and aside from adding structural information predicted by OmegaFold (while AlphaFold-predicted structures for DMS data are also available on the ProteinGym website), we did not perform any additional preprocessing. Therefore, we do not release a separate preprocessed version of the data.

• **Usage**. In the zero-shot setting, this evaluation involves no training process. Instead, it directly infers and quantifies the log-likelihood of protein variants under both sequence and structural context.

• **License**. The original dataset, ProteinGym, is available under a MIT license at `https://github.com/OATML-Markslab/ProteinGym/blob/main/LICENSE`.

## E  DOMAIN-SPECIFIC MODEL ARCHITECTURE

**KDBNet**. KDBNet (Luo et al., 2023b) is a graph neural network model designed for predicting kinase–small molecule binding affinity. It constructs heterogeneous graph pairs from the 3D structures of protein binding pockets and small molecules, and employs two structure-aware GNNs to

learn their respective representations. For the protein, the model focuses on 85 binding site residues defined by the KLIFS database to build a structure graph $\mathcal{G}_p = (\mathcal{V} * p, \mathcal{E} * p)$, where nodes represent residues and edges are formed between C$*\alpha$ atoms within 8 Å. Each residue node includes three types of features: one-hot encoding of amino acid type, geometric features based on backbone conformation (e.g., $(\sin\phi_i, \sin\psi_i, \sin\omega_i, \cos\phi_i, \cos\psi_i, \cos\omega_i)$), and evolutionary embeddings obtained from the ESM language model. Edge features consist of four components: radial basis function (RBF) encoding of distances, local frame-based projections of relative direction vectors, rotational quaternions between residues, and relative position encodings using a Transformer-based function $E * \text{pos}(c_j - c_i)$. The small molecule is represented as a graph $\mathcal{G}_d = (\mathcal{V}_d, \mathcal{E}_d)$, where atoms are nodes and edges connect atom pairs within 4.5 Å. Each atom node includes both 3D coordinates and a 66-dimensional scalar feature vector describing chemical properties, while edge features include unit directional vectors and RBF-encoded distances. For encoding protein structure, KDB-Net uses a Graph Transformer, where each layer updates node features via the following attention mechanism:

$$h_i^{(\ell)} = W_1^{(\ell)} h_i^{(\ell-1)} + \sum_{j \in \mathcal{N}(i)} \alpha_{ij} \left( W_2^{(\ell)} h_j^{(\ell-1)} + W_3^{(\ell)} e_{ij} \right). \tag{1}$$

With attention weights defined as:

$$\alpha_{ij} = \text{softmax} \left( \frac{\left( W_4^{(\ell)} h_i^{(\ell-1)} \right)^\top \left( W_5^{(\ell)} h_j^{(\ell-1)} + W_3^{(\ell)} e_{ij} \right)}{\sqrt{d_\ell}} \right). \tag{2}$$

Three such graph convolution layers are stacked with Leaky ReLU activations, and global sum pooling is applied to obtain the final protein embedding. For small molecules, KDBNet adopts the GVP-GNN architecture, which jointly models vector and scalar features to maintain rotational and translational equivariance. Each atom node is represented as a tuple $(v_i^v, v_i^s)$ and each edge as $(e_{ij}^v, e_{ij}^s)$, where the vector part allows direct alignment with atomic coordinates. The final protein and drug embeddings are passed through fully connected layers to regress the binding affinity.

**DeepPROTACs**. DeepPROTAC (Li et al., 2022) is a deep learning framework designed to predict the degradation efficacy of PROTAC molecules by integrating structural and chemical information from multiple components: the protein of interest (POI), the E3 ligase, and the PROTAC compound itself. The input includes five parts: the POI binding pocket, the E3 ligase binding pocket, the warhead (the PROTAC moiety binding to the POI), the E3 ligand (binding to the E3 ligase), and the linker represented as a SMILES string. For the four molecular structures (POI pocket, E3 pocket, warhead, and E3 ligand), atom-level graphs are constructed and processed using Graph Convolutional Layers (GCLs) followed by max pooling to obtain fixed-size feature embeddings. The linker is embedded using an LSTM network to capture sequential chemical features. All five embeddings are concatenated and passed through fully connected layers to yield a binary output: 1 indicates good degradation (defined as $\text{DC}_{50} \leq 100\,\text{nM}$ and $\text{D}_{max} \geq 80\%$), while 0 indicates poor degradation ($\text{DC}_{50} > 100\,\text{nM}$ or $\text{D}_{max} < 80\%$). This modular architecture enables DeepPROTACs to effectively model ternary complex formation and predict degradation outcomes based on both structural and sequential representations.

**ET-PROTACs**. ET-PROTACs (Cai et al., 2025) is an end-to-end deep learning model designed for PROTAC degradation prediction, consisting of four main components: (i) initial PROTAC and protein featurization; (ii) representation learning using 3D graph-based and sequence-based encoders; (iii) a cross-modal ternary attention block; and (iv) a final classifier. Each component is described in detail below. Each PROTAC is represented as a 2D molecular graph $\mathcal{G} = (\mathcal{V}, \mathcal{E})$ and associated with a 3D coordinate matrix $X \in \mathbb{R}^{|\mathcal{V}| \times 3}$ using RDKit[2]. Nodes $v_i \in \mathcal{V}$ represent atoms, and edges $a_{ij} \in \mathcal{E}$ denote chemical bonds. Node features $h_i$ and edge features $a_{ij}$ are encoded using learned embeddings. Additionally, atomic 3D coordinates $x_i \in \mathbb{R}^3$ are embedded to capture spatial information. The input protein is given as a sequence $P = \{a_1, a_2, \ldots, a_n\}$ of 23 amino acids (including one non-standard residue). Protein sequences are encoded by two embedding layers: a learned character embedding of dimension 64, and a positional embedding of the same dimension.

---

[2]https://www.rdkit.org/

The molecular graph and 3D coordinates are processed by an Equivariant Graph Neural Network (EGNN), which is invariant to rotations and translations. Let $h_i^\ell$, $x_i^\ell$, and $a_{ij}$ be the node features, coordinates, and edge features at layer $\ell$, respectively. The EGNN updates are as follows:

$$
\begin{aligned}
m_{ij} &= \phi_e(h_i^\ell, h_j^\ell, \|x_i^\ell - x_j^\ell\|^2, a_{ij}), \\
x_i^{\ell+1} &= x_i^\ell + C \sum_{j \neq i} (x_i^\ell - x_j^\ell)\phi_x(m_{ij}), \\
m_i &= \sum_{j \in \mathcal{N}(i)} m_{ij}, \\
h_i^{\ell+1} &= \phi_h(h_i^\ell, m_i),
\end{aligned}
\tag{1}
$$

where $C = 1/(M-1)$, $\phi_e$, $\phi_x$, and $\phi_h$ are learnable functions, and $M$ is the number of atoms. Final PROTAC embeddings are obtained by combining features of the warhead, linker, and E3 ligase substructures. ET-PROTACs leverages CNNs to encode protein sequences directly from FASTA format. The sequence embedding $E$ is passed through three CNN layers to yield latent protein features $P_{\text{cnn}} \in \mathbb{R}^{d \times f}$, where $f$ is the number of filters in the last layer. Given feature matrices $D = \{d_1, \ldots, d_X\}$ (PROTAC), $P = \{p_1, \ldots, p_Y\}$ (protein), and $L = \{l_1, \ldots, l_Z\}$ (ligase), we form composite pairs:

$$
\begin{aligned}
\text{pd}_i &= \text{CONCAT}(p_j, d_i), \\
\text{dl}_j &= \text{CONCAT}(d_i, l_k), \\
\text{pda}_i &= F(W_{\text{pd}} \cdot \text{pd}_i + b), \\
\text{dla}_j &= F(W_{\text{dl}} \cdot \text{dl}_j + b),
\end{aligned}
\tag{2}
$$

where $F$ is a non-linear activation, and $W_{\text{pd}}, W_{\text{dl}} \in \mathbb{R}^{f \times f}$ are learned weights. The combined attention is computed as:

$$
\begin{aligned}
A_{i,j} &= F(W_a \cdot (\text{pda}_i + \text{dla}_j) + b), \\
A_{\text{pd}} &= \sigma\left(\text{MEAN}(A, 2)\right), \\
A_{\text{dl}} &= \sigma\left(\text{MEAN}(A, 1)\right),
\end{aligned}
\tag{3}
$$

where $\sigma$ denotes the sigmoid function. The resulting outputs $V_{\text{pd}}$ and $V_{\text{dl}}$ are the attention-enriched representations of PROTAC–protein and PROTAC–ligase interactions. The attention outputs are pooled over sequence length and concatenated:

$$
s = \text{LeakyReLU}(W_s \cdot \text{CONCAT}(V_{\text{pd}}, V_{\text{dl}}) + b_s),
\tag{4}
$$

where $W_s$ is a learnable weight matrix. Dropout is applied before and after each linear layer to prevent overfitting.

**DeepFRI**. DeepFRI (Gligorijević et al., 2021) is a deep learning framework for protein function prediction based solely on amino acid sequences. Given a protein sequence of length $L$, the input is encoded as a binary matrix $X = [x_1, \ldots, x_L] \in \{0,1\}^{L \times 26}$, where each $x_i$ is a one-hot vector indicating the amino acid type at position $i$, including 20 standard residues, 5 non-standard residues, and a gap symbol. This input is passed through a set of one-dimensional convolutional layers, each consisting of $f_n = 512$ filters of varying kernel lengths $f_l$, followed by rectified linear unit (ReLU) activation, $\text{ReLU}(x) = \max(x, 0)$, and a global max-pooling operation. The use of multiple filter sizes enables the extraction of complementary local patterns from the sequence. Outputs from 16 such CNN layers are concatenated, resulting in an $L \times 8192$ feature representation. Finally, a fully connected layer with sigmoid activation outputs probabilities for either Gene Ontology (GO) terms or Enzyme Commission (EC) classes. The output dimensionality is task-specific, corresponding to $|\text{GO}|$ or $|\text{EC}|$ respectively.

**DPFunc**. DPFunc (Wang et al., 2025) is a structure-aware protein function prediction framework that captures residue-level information by integrating 3D geometric structure and pretrained sequence embeddings. For a given protein of length $l$, a residue-level undirected graph is constructed where each node represents a residue, and an edge is added between two residues if the distance between their $C_\alpha$ atoms is less than 10 Å. This defines the adjacency matrix $A \in \{0,1\}^{l \times l}$. Node features are initialized using embeddings from a pretrained protein language model (ESM-1b), resulting in $X \in \mathbb{R}^{l \times d}$. Two graph convolutional layers are then used to propagate structural information, with residual connections added to facilitate gradient flow. The update rule for the $k$-th GCN

layer is given by:

$$X^{(k+1)} = X^{(k)} + \text{ReLU}\left(\tilde{D}^{-1/2}\tilde{A}\tilde{D}^{-1/2}X^{(k)}W^{(k)}\right), \tag{1}$$

where $\tilde{A} = A + I$ adds self-loops and $\tilde{D}$ is its corresponding degree matrix. After GCN propagation, domain-level information is introduced to enhance residue representations. Protein domains are annotated using InterProScan, and their one-hot encoding $IPR \in \{0,1\}^{1\times m}$ is mapped to dense embeddings via two fully connected layers with ReLU activations:

$$H = \text{ReLU}\left((\text{ReLU}(IPR \cdot W_{\text{emd}})W_1 + b_1)W_2 + b_2\right). \tag{2}$$

A domain-guided attention mechanism, inspired by the Transformer encoder, is applied to model the interaction between domain and residue representations. For each attention head $i$, the query, key, and value matrices are computed as:

$$Q_i = HW_i^Q, \quad K_i = X_{\text{final}}W_i^K, \quad V_i = X_{\text{final}}W_i^V, \tag{3}$$

and attention weights are derived as:

$$W_i^A = \text{Softmax}\left(\frac{K_i Q_i^\top}{\sqrt{d}}\right). \tag{4}$$

The multi-head attention output is computed as:

$$X_{\text{multi}} = \text{LayerNorm}\left(\text{Concat}(W_1^A V_1, \ldots, W_n^A V_n)W_{\text{trans}} + X_{\text{final}}\right), \tag{4}$$

followed by a feedforward transformation with residual connection:

$$X_{\text{out}} = \text{LayerNorm}\left(\text{FF}(X_{\text{multi}}) + X_{\text{multi}}\right). \tag{5}$$

Protein-level features are then obtained by summing across residues:

$$x_{\text{pool}} = \sum_{i=1}^{l} X_{\text{out}}[i]. \tag{6}$$

This representation is concatenated with the average of initial residue features:

$$x_{\text{integrate}} = \text{Concat}\left(x_{\text{pool}}, \frac{1}{l}\sum_{i=1}^{l} X[i]\right), \tag{7}$$

and passed through a multilayer perceptron with sigmoid activation to produce GO term probabilities:

$$\hat{y} = \text{Sigmoid}(\text{MLP}(x_{\text{integrate}})). \tag{8}$$

To ensure consistency with the GO hierarchy, a postprocessing step enforces that if a child term is predicted, all its ancestors are also predicted:

$$\hat{y}_i^{\text{post}} = \max\left(\hat{y}_i, \hat{y}_{\text{child}_1}, \ldots, \hat{y}_{\text{child}_n}\right). \tag{9}$$

This hierarchical correction is applied only after inference, without affecting training efficiency.

**UniZyme**. UniZyme (Li et al., 2025) is a biochemically informed framework designed to generalize protein cleavage site prediction across diverse enzymes. It comprises an enzyme encoder and a substrate encoder. The enzyme encoder integrates sequence-derived features with energetic frustration and 3D structural information. Given an enzyme $P_e = (X, R)$, where $X$ are residue embeddings and $R$ are C$^\alpha$ coordinates, the pairwise frustration score is defined as:

$$F(i,j) = \frac{E(i,j) - \mu_{\text{rand}}(i,j)}{\sigma_{\text{rand}}(i,j)}. \tag{1}$$

To incorporate spatial and energetic cues into the self-attention mechanism, each residue pair is encoded via Gaussian basis kernels:

$$\Phi_{i,j}^{\text{energy}} = \text{MLP}(\phi_{\text{energy}}(F(i,j))), \quad \Phi_{i,j}^{\text{dist}} = \text{MLP}(\phi_{\text{dist}}(\|r_i - r_j\|_2)). \tag{2}$$

These terms are added to the attention score matrix in the graph transformer:

$$A_{i,j}^k = \frac{(h_i^{k-1} W_Q)(h_j^{k-1} W_K)^T}{\sqrt{d}} + \Phi_{i,j}^{\text{energy}} + \Phi_{i,j}^{\text{dist}}. \tag{3}$$

To guide the encoder toward catalytically relevant regions, an auxiliary active-site prediction is introduced:

$$\hat{a}_i = \sigma(h_i \cdot w_a). \tag{4}$$

The predicted probabilities $\hat{a}_i$ are then used in a soft attention-like pooling mechanism for enzyme representation:

$$h_e = \sum_{i=1}^{N} \frac{f(\hat{a}_i)}{\sum_j f(\hat{a}_j)} h_i. \tag{5}$$

For a substrate $P_s$, cleavage site prediction is formulated by concatenating local substrate representations $H_s^{t:t+l}$ with the enzyme representation $h_e$:

$$\hat{c}_{e,s}^{(t)} = \text{MLP}(\text{CONCAT}(H_s^{t:t+l}, h_e)). \tag{6}$$

The model is jointly optimized using binary cross-entropy losses for both tasks:

$$\mathcal{L} = \mathcal{L}_c(D_c) + \lambda \mathcal{L}_a(D_a). \tag{7}$$

**MONN**. MONN (Li et al., 2020b) is composed of four interconnected modules: a graph convolutional module for molecular representation, a CNN module for residue-level protein representation, a pairwise interaction module for atom-residue interaction estimation, and an affinity prediction module for compound-protein binding affinity estimation.

Given a molecular graph $G = (V, E)$, each atom $v_i \in V$ is initially represented by an 82-dimensional one-hot vector $v_i^{\text{init}}$ encoding atomic features. These are projected into a hidden space $\mathbb{R}^{h_1}$ by:

$$v_i^0 = f(W_{\text{init}} v_i^{\text{init}}), \tag{1}$$

where $f(x) = \max(0, x) + 0.1 \min(0, x)$ is the leaky ReLU activation, and $W_{\text{init}} \in \mathbb{R}^{h_1 \times 82}$. Bonds $e_{i,j} \in E$ are represented by 6-dimensional one-hot vectors encoding bond type and topology.

For $L$ graph convolutional iterations, features are updated by message passing and graph warp. At each layer $l$, local messages are aggregated:

$$t_i^l = \sum_{v_k \in \mathcal{N}(v_i)} f(W_{\text{gather}}^l [v_k^{l-1}, e_{i,k}]), \tag{2}$$

followed by feature updates:

$$u_i^l = f(W_{\text{update}}^l [t_i^l, v_i^{l-1}]). \tag{3}$$

Global information is captured via a super node $s^l$ that interacts with all $u_i^l$ to yield the final atom features $\{v_i^L\}$ and compound feature $s^L$.

Protein sequences are encoded by mapping residues to BLOSUM62 (Eddy, 2004) columns and processed through a 1D CNN to obtain residue embeddings $\{r_j\}$ in $\mathbb{R}^{h_1}$.

Atom-residue interactions are predicted by projecting atom and residue embeddings to a shared space and computing their dot-product, followed by a sigmoid:

$$P_{i,j} = \sigma(f(W_{\text{atom}} v_i) \cdot f(W_{\text{residue}} r_j)). \tag{4}$$

For affinity prediction, atom, residue, and super node features are transformed via:

$$h_{v,i} = f(W_v v_i), \quad h_s = f(W_s s), \quad h_{r,j} = f(W_r r_j), \tag{5}$$

where $W_v, W_r, W_s \in \mathbb{R}^{h_2 \times h_1}$. A modified dual attention network uses the interaction matrix $P$ to compute attentions $\{\alpha_{v,i}\}, \{\alpha_{r,j}\}$, aggregating compound and protein representations as:

$$h_c = \sum_i \alpha_{v,i} h_{v,i}, \quad h_p = \sum_j \alpha_{r,j} h_{r,j}. \tag{6}$$

The binding affinity is predicted by a regression on the outer product between $[h_c, h_s]$ and $h_p$:

$$a = W_{\text{affinity}} f(\text{flatten}([h_c, h_s] \otimes h_p)), \tag{7}$$

where $W_{\text{affinity}} \in \mathbb{R}^{1 \times 2h_2^2}$.

**CLIPZyme**. CLIPZyme (Mikhael et al., 2024) formulates enzyme screening as a retrieval task, where a predefined set of enzymes is ranked based on their predicted ability to catalyze a given chemical reaction. Each reaction $R$ and enzyme $P$ is encoded into a $d$-dimensional vector $r, p \in \mathbb{R}^d$ using learned encoders $f_{\text{rxn}}$ and $f_{\text{p}}$, respectively. A scoring function $s(r, p)$ computes the cosine similarity between the two embeddings:

$$s_{ij} = s(r_i, p_j) = \frac{r_i \cdot p_j}{\|r_i\| \|p_j\|}. \tag{1}$$

To align reactions with their catalyzing enzymes, a symmetric contrastive loss is used:

$$\mathcal{L}_{ij} = -\frac{1}{2N} \left( \log \frac{e^{s_{ij}/\tau}}{\sum_k e^{s_{ik}/\tau}} + \log \frac{e^{s_{ij}/\tau}}{\sum_k e^{s_{kj}/\tau}} \right), \tag{2}$$

where $\tau$ is a temperature parameter and negative samples are drawn from other enzymes in the batch.

To represent a reaction, atom-mapped molecular graphs of the reactants $G_x$ and products $G_y$ are first encoded by a directed message passing neural network (Yang et al., 2019) (DMPNN) $f_{\text{mol}}$, yielding atom features $a_i$ and bond features $b_{ij}$:

$$a_i, b_{ij} = f_{\text{mol}}(G_x, G_y). \tag{3}$$

A pseudo-transition state graph $G_{TS}$ is then constructed using shared atom features and summed bond features:

$$v_i^{(TS)} = v_i^{(x)} = v_i^{(y)}, \tag{3}$$

$$e_{ij}^{(TS)} = b_{ij}^{(x)} + b_{ij}^{(y)}. \tag{4}$$

This graph is processed by a second DMPNN $f_{TS}$, and the reaction embedding is obtained by aggregating the learned node features:

$$a_i', b_{ij}' = f_{TS}(G_{TS}), \tag{4}$$

$$r = \sum_i a_i'. \tag{5}$$

Each protein is modeled as a 3D graph $G_p = (V, E)$ with node features $h_i$, edge features $e_{ij}$, and 3D coordinates $c_i \in \mathbb{R}^3$. Residue features are initialized using ESM-2 (650M) embeddings with dimensionality 1280. An EGNN with coordinate updates is used to encode $G_p$ into the final protein embedding $p = f_{\text{p}}(G_p)$. Relative distances between residues are encoded using a sinusoidal basis to enhance structural modeling.

# F PRETRAIN MODEL ARCHITECTURE AND EXPERIMENTAL SETUP

## F.1 MODEL ARCHITECTURE

This section details the models employed in this study.

**Equivariant Graph Neural Networks (EGNN)**. EGNN (Satorras et al., 2021) is designed to process graphs where each node is associated with both feature embeddings and spatial coordinates. EGNN preserves equivariance under Euclidean transformations (translation and rotation) and node permutations, making it well-suited for modeling molecular and protein structures.

Given a graph $\mathcal{G} = (\mathcal{V}, \mathcal{E})$, each node $v_i \in \mathcal{V}$ has a feature vector $\mathbf{h}_i \in \mathbb{R}^{d_h}$ and a coordinate $x_i \in \mathbb{R}^n$. The Equivariant Graph Convolutional Layer (EGCL) updates node features and coordinates as follows:

$$\mathbf{m}_{ij} = \phi_e\left(\mathbf{h}_i^l, \mathbf{h}_j^l, \|\mathbf{x}_i - \mathbf{x}_j\|^2, a_{ij}\right), \tag{1}$$

$$\mathbf{x}_i^{l+1} = \mathbf{x}_i^l + C \sum_{j \neq i} (\mathbf{x}_i^l - \mathbf{x}_j^l) \cdot \phi_x(\mathbf{m}_{ij}), \tag{2}$$

$$\mathbf{m}_i = \sum_{j \neq i} \mathbf{m}_{ij}, \tag{3}$$

$$\mathbf{h}_i^{l+1} = \phi_h(\mathbf{h}_i^l, \mathbf{m}_i), \tag{4}$$

where $\phi_e$ is an edge function that computes a message embedding from node features, squared distance, and optional edge attributes $a_{ij}$. $\phi_x$ outputs a scalar weight for coordinate updates, based on $m_{ij}$. $\phi_h$ is a node update function. $C$ is a normalization constant, typically $C = 1/(|\mathcal{V}| - 1)$.

Eq. (2) ensures that the coordinate update is equivariant to rotations and translations by acting as a learnable radial vector field. EGNN combines geometric awareness with standard message passing, making it a powerful architecture for modeling 3D protein structures.

**SE(3) Transformer**. The SE(3) Transformer (Fuchs et al., 2020) is a neural architecture designed to model geometric data such as molecules and point clouds while respecting SE(3) symmetries, i.e., 3D rotations and translations. It achieves this through the integration of *Tensor Field Networks (TFNs)* (Thomas et al., 2018) for equivariant message passing and *invariant attention mechanisms* for weighted aggregation. This section presents a unified formulation of the SE(3) Transformer, encompassing both TFN convolution and attention-based update steps.

Formally, each node $i$ in the graph is associated with a position $\mathbf{x}_i \in \mathbb{R}^3$ and a set of typed features $\mathbf{f}_i = \bigoplus_{\ell \geq 0} \mathbf{f}_i^\ell$, where $\mathbf{f}_i^\ell \in \mathbb{R}^{(2\ell+1) \times d}$ denotes a rank-$\ell$ feature tensor (type-$\ell$ representation of SO(3)) with $d$ channels.

The output features at node $i$ and type $\ell$ are computed as:

$$\mathbf{f}_{\text{out},i}^\ell = \mathbf{W}_{\text{self}}^{\ell\ell} \mathbf{f}_{\text{in},i}^\ell + \sum_{k \geq 0} \sum_{j \in \mathcal{N}_i \setminus \{i\}} \alpha_{ij} \mathbf{W}^{\ell k}(\mathbf{x}_j - \mathbf{x}_i) \mathbf{f}_{\text{in},j}^k. \tag{1}$$

The first term is a type-preserving self-interaction, and the second term aggregates messages from neighbors $j \in \mathcal{N}_i$ via a convolution-like operation using TFN kernels $\mathbf{W}^{\ell k}(\mathbf{x}_j - \mathbf{x}_i)$, modulated by scalar attention weights $\alpha_{ij} \in \mathbb{R}$.

The kernel $\mathbf{W}^{\ell k}: \mathbb{R}^3 \to \mathbb{R}^{(2\ell+1) \times (2k+1)}$ is constructed to ensure SE(3)-equivariance, and is defined as a linear combination of spherical harmonic projections:

$$\mathbf{W}^{\ell k}(\mathbf{x}) = \sum_{J=|\ell-k|}^{\ell+k} \varphi_J^{\ell k}(\|\mathbf{x}\|) \sum_{m=-J}^{J} Y_{Jm}(\widehat{\mathbf{x}}) \, \mathbf{Q}_{Jm}^{\ell k}, \tag{2}$$

where $\widehat{\mathbf{x}} = \mathbf{x}/\|\mathbf{x}\|$, $Y_{Jm}$ is the $m$-th spherical harmonic of order $J$, $\varphi_J^{\ell k}$ is a learnable radial function, and $\mathbf{Q}_{Jm}^{\ell k}$ are learnable matrices formed from Clebsch–Gordan coefficients. This construction guarantees equivariance under SE(3) transformations by disentangling radial and angular dependencies.

The attention weights $\alpha_{ij}$ are designed to be SE(3)-invariant and are computed via a dot-product attention mechanism:

$$\alpha_{ij} = \frac{\exp(\langle \mathbf{q}_i, \mathbf{k}_{ij} \rangle)}{\sum_{j' \in \mathcal{N}_i \setminus \{i\}} \exp(\langle \mathbf{q}_i, \mathbf{k}_{ij'} \rangle)}, \tag{3}$$

where the query vector $\mathbf{q}_i$ and key vector $\mathbf{k}_{ij}$ are both constructed from the input features through learned TFN mappings:

$$\mathbf{q}_i = \bigoplus_{\ell \geq 0} \sum_{k \geq 0} \mathbf{W}_Q^{\ell k} \mathbf{f}_{\text{in},i}^k, \quad \mathbf{k}_{ij} = \bigoplus_{\ell \geq 0} \sum_{k \geq 0} \mathbf{W}_K^{\ell k}(\mathbf{x}_j - \mathbf{x}_i) \mathbf{f}_{\text{in},j}^k. \tag{4}$$

Here, $\mathbf{W}_Q^{\ell k}$ and $\mathbf{W}_K^{\ell k}$ are TFN-type filters that output representations in the same basis, ensuring that the dot product is invariant under common SO(3) actions.

Equivariance is preserved in the message-passing path through the linear combination of TFN kernels. The attention mechanism maintains invariance because it operates entirely in scalar (dot-product) space between representations of the same type, which are invariant under group actions due to orthonormality properties of spherical harmonics.

**GVP**. The Geometric Vector Perceptron (GVP) (Jing et al., 2021) is a neural module designed for learning over geometric data, in which each entity (e.g., an amino acid residue or atom) is represented by both scalar and vector features. Formally, given a pair $(\mathbf{s}, \mathbf{V})$ where $\mathbf{s} \in \mathbb{R}^n$ denotes scalar features and $\mathbf{V} \in \mathbb{R}^{\nu \times 3}$ denotes geometric vector features, the GVP outputs a new pair $(\mathbf{s}', \mathbf{V}') \in \mathbb{R}^m \times \mathbb{R}^{\mu \times 3}$. This transformation is designed to ensure that the scalar outputs remain invariant while the vector outputs are equivariant with respect to rotations and reflections in $\mathbb{R}^3$.

The GVP proceeds via the following algorithm:

---

**Algorithm 1** Geometric Vector Perceptron (GVP)

---

**Input:** Scalar features $\mathbf{s} \in \mathbb{R}^n$, vector features $\mathbf{V} \in \mathbb{R}^{\nu \times 3}$
**Output:** Updated features $(\mathbf{s}', \mathbf{V}') \in \mathbb{R}^m \times \mathbb{R}^{\mu \times 3}$

  1: $\mathbf{V}_h \leftarrow \mathbf{W}_h \mathbf{V}$
  2: $\mathbf{V}_\mu \leftarrow \mathbf{W}_\mu \mathbf{V}_h$
  3: $\mathbf{s}_h \leftarrow \|\mathbf{V}_h\|_2$ (row-wise)
  4: $\mathbf{v}_\mu \leftarrow \|\mathbf{V}_\mu\|_2$ (row-wise)
  5: $\mathbf{s}_{h+n} \leftarrow \texttt{concat}(\mathbf{s}_h, \mathbf{s})$
  6: $\mathbf{s}_m \leftarrow \mathbf{W}_m \mathbf{s}_{h+n} + \mathbf{b}$
  7: $\mathbf{s}' \leftarrow \sigma(\mathbf{s}_m)$
  8: $\mathbf{V}' \leftarrow \sigma_+(\mathbf{v}_\mu) \odot \mathbf{V}_\mu$
  9: **return** $(\mathbf{s}', \mathbf{V}')$

---

Here, $\mathbf{W}_h$, $\mathbf{W}_\mu$, and $\mathbf{W}_m$ are learnable linear transformations, while $\sigma$ and $\sigma_+$ are nonlinear activation functions (e.g., ReLU, GELU, or their variants). The vector norm computations $\| \cdot \|_2$ are row-wise and used to extract invariant scalar information from the geometric vectors, which is then injected into the scalar pathway before transformation. The final output $\mathbf{V}'$ is modulated via element-wise multiplication with a positive gating function $\sigma_+(\mathbf{v}_\mu)$ to preserve equivariance.

The GVP-GNN updates node embeddings $h_v^{(i)}$ via message passing:

$$h_m^{(j \to i)} = \text{GVP}\left(\texttt{concat}(h_v^{(j)}, h_e^{(j \to i)})\right), \tag{1}$$

$$h_v^{(i)} \leftarrow \text{LayerNorm}\left(h_v^{(i)} + \frac{1}{k'} \sum_{j \in \mathcal{N}_i} \text{Dropout}(h_m^{(j \to i)})\right), \tag{2}$$

Here, $\text{GVP}(\cdot)$ denotes a sequence of three GVPs. The embeddings $h_v^{(i)}$ and $h_e^{(j \to i)}$ correspond to node $i$ and edge $(j \to i)$, respectively, as previously defined. The message $h_m^{(j \to i)}$ is computed from these embeddings and represents the information passed from node $j$ to node $i$. The variable $k'$ denotes the number of incoming messages, which equals $k$ unless the protein contains fewer than $k$ amino acid residues. Then, a feed-forward point-wise layer is utilized to update the node embeddings at all nodes $i$:

$$h_v^{(i)} \leftarrow \text{LayerNorm}\left(h_v^{(i)} + \text{Dropout}(\text{GVP}(h_v^{(i)}))\right), \tag{3}$$

where $\text{GVP}(\cdot)$ is a sequence of two GVPs. This architecture allows for expressive, symmetry-aware modeling of protein geometry with built-in equivariance, while remaining efficient and conceptually simple.

**ProteinBERT**. ProteinBERT (Brandes et al., 2022) is a denoising autoencoder for proteins, inspired by BERT (Devlin et al., 2019) but with a distinct architecture. It takes two inputs: amino acid sequences and GO annotations. The architecture consists of parallel local and global pathways. Local

representations are 3D tensors of shape $B \times L \times d_{\text{local}}$ (with $d_{\text{local}} = 128$), and global representations are 2D tensors of shape $B \times d_{\text{global}}$ (with $d_{\text{global}} = 512$).

Each input sequence is embedded into local features by a shared position-wise embedding layer, while annotations are mapped into global features via a fully connected layer. The model applies six transformer-like blocks (Vaswani et al., 2017), each updating local features using both narrow and dilated convolutions (kernel size 9, dilation 1 and 5), followed by feedforward layers. The global path consists of two fully connected layers per block.

Local-global interaction occurs via (1) a broadcast fully connected layer from global to local, and (2) a global attention layer from local to global. Global attention has linear complexity and uses trainable projection matrices $W_q$, $W_k$, and $W_v$ (with $d_{\text{key}} = 64$, $d_{\text{value}} = 128$), and $n_{\text{heads}} = 4$ per block. All activations use GELU (Hendrycks & Gimpel, 2016).

**D-Transformer.** D-Transformer augments the Transformer with 3D knowledge by adding pairwise $C\alpha$ distances to the self-attention scores, similar to Transformer-M (Luo et al., 2023a). For a protein of length $L$, let $\mathbf{X} = [\mathbf{x}_1, \ldots, \mathbf{x}_L] \in \mathbb{R}^{L \times d}$ be residue embeddings and $D(i,j) = \|\mathbf{r}_i - \mathbf{r}_j\|_2$ the Euclidean distance between residues $i$ and $j$.

Each distance is first expanded with a learnable Gaussian radial basis:

$$\phi_{\text{dist}}(D(i,j)) = \left[ \exp\left( -\frac{(D(i,j) - \mu_k)^2}{2\sigma_k^2} \right) \right]_{k=1}^{K}, \tag{1}$$

where $\{\mu_k, \sigma_k\}_{k=1}^{K}$ are trainable parameters. A multilayer perceptron maps this $K$-vector to a scalar bias $b_{ij} = \text{MLP}(\phi_{\text{dist}}(D(i,j)))$.

For one attention head, the unnormalised score is

$$A_{ij} = \frac{(\mathbf{h}_i W_Q)(\mathbf{h}_j W_K)^\top}{\sqrt{d}} + b_{ij}, \tag{2}$$

with $\mathbf{h}_i$ the hidden state at residue $i$ and learned $W_Q, W_K \in \mathbb{R}^{d \times d}$. Because $b_{ij}$ depends only on distances, $A_{ij}$ is invariant to global rotations, translations, and residue permutations. Normalised weights and value aggregation follow standard Transformer rules:

$$\alpha_{ij} = \text{softmax}_j(A_{ij}), \qquad \mathbf{z}_i = \sum_{j=1}^{L} \alpha_{ij}(\mathbf{h}_j W_V), \tag{3}$$

where $W_V \in \mathbb{R}^{d \times d}$. A position-wise feed-forward network with residual connections completes each layer, and stacking multiple such layers yields a lightweight architecture that leverages structural information without coordinate updates.

**ESM2**. ESM-2 (Lin et al., 2023) is a family of transformer-based protein language models trained to predict masked amino acids in protein sequences. Compared to its predecessor ESM-1b (Rives et al., 2021), ESM-2 introduces architectural refinements, improved training procedures, and is scaled across model sizes ranging from 8M to 15B parameters. The model is trained using a standard masked language modeling (MLM) objective: for 15% randomly masked positions in a sequence, the model predicts the identity of each masked amino acid based on its unmasked context.

Formally, the training objective is:

$$\mathcal{L}_{\text{MLM}} = \sum_{i \in M} \log p(x_i \mid x_{\backslash M}), \tag{1}$$

where $M$ is the set of masked positions and $x_{\backslash M}$ denotes the observed amino acids. Training is performed on $\sim$65 million sequences sampled from $\sim$43 million UniRef50 (Consortium, 2019) clusters, covering $\sim$138 million UniRef90 entries.

Despite its unsupervised formulation, ESM-2 learns rich structural features purely from sequence data. The model achieves state-of-the-art performance on several protein structure prediction benchmarks and surpasses previous models, e.g., ESM-1b, ProteinBERT.

Table 13: Hyperparameter setting of the EGNN pre-training.

| MLM | |
| --- | --- |
| Batch Size | 48 |
| Learning Rate | 1e-3 |
| Warmup Ratio | 0.05 |
| Mask Ratio | 0.15 |
| Hidden Dimension | 512 |
| Layer Depth | 3 |
| MVCL | |
| Batch Size | 96 |
| Learning Rate | 1e-3 |
| Warmup Ratio | 0.05 |
| Subsequence Length | 50 |
| Max Node | 50 |
| Temperature | 0.0099 |
| Hidden Dimension | 512 |
| Layer Depth | 3 |
| PFP | |
| Batch Size | 48 |
| Learning Rate | 1e-3 |
| Warmup Ratio | 0.05 |
| Temperature | 0.01 |
| Hidden Dimension | 512 |
| Layer Depth | 3 |

Table 14: Hyperparameter setting of the SE(3) Transformer pre-training.

| MLM | |
| --- | --- |
| Batch Size | 48 |
| Learning Rate | 1e-2 |
| Warmup Ratio | 0.05 |
| Mask Ratio | 0.15 |
| Hidden Dimension | 36 |
| Layer Depth | 2 |
| MVCL | |
| Batch Size | 96 |
| Learning Rate | 1e-2 |
| Warmup Ratio | 0.05 |
| Subsequence Length | 50 |
| Max Node | 50 |
| Temperature | 0.0099 |
| Hidden Dimension | 36 |
| Layer Depth | 2 |
| PFP | |
| Batch Size | 48 |
| Learning Rate | 1e-3 |
| Warmup Ratio | 0.05 |
| Temperature | 0.0099 |
| Hidden Dimension | 36 |
| Layer Number | 2 |

## F.2 PRETRAIN MODEL EXPERIMENTAL SETUP

In this section, we provide more details about the pertaining.

**Pre-training Data.** The structural information used for pretraining is derived from the AFDB Swiss-Prot dataset, while the functional and family annotations are obtained from UniProt Swiss-Prot.

Table 15: Hyperparameter setting of the D-Transformer pre-training.

| MLM | |
|---|---|
| Batch Size | 16 |
| Learning Rate | 1e-4 |
| Warmup Ratio | 0.1 |
| Mask Ratio | 0.15 |
| Hidden Dimension | 256 |
| Layer Depth | 6 |
| MVCL | |
| Batch Size | 16 |
| Learning Rate | 1e-4 |
| Warmup Ratio | 0.1 |
| Subsequence Length | 50 |
| Max Node | 50 |
| Temperature | 0.01 |
| Hidden Dimension | 256 |
| Layer Depth | 6 |
| PFP | |
| Batch Size | 16 |
| Learning Rate | 1e-4 |
| Warmup Ratio | 0.1 |
| Temperature | 0.01 |
| Hidden Dimension | 256 |
| Layer Number | 6 |

Table 16: Hyperparameter setting of the GVP pre-training.

| MLM | |
|---|---|
| Batch Size | 64 |
| Learning Rate | 1e-4 |
| Warmup Ratio | 0.1 |
| Mask Ratio | 0.15 |
| Node Dimension | (155,16) |
| Layer Depth | 3 |
| MVCL | |
| Batch Size | 64 |
| Learning Rate | 1e-4 |
| Warmup Ratio | 0.1 |
| Subsequence Length | 50 |
| Max Node | 50 |
| Temperature | 0.01 |
| Node Dimension | (155,16) |
| Layer Depth | 3 |
| PFP | |
| Batch Size | 64 |
| Learning Rate | 1e-4 |
| Warmup Ratio | 0.1 |
| Temperature | 0.01 |
| Node Dimension | (155,16) |
| Layer Number | 3 |

Swiss-Prot is a high-quality, manually curated protein sequence database within UniProt. Its goal is to provide comprehensive and biologically meaningful annotations of protein sequences, such as their functions, families, and maintain minimal redundancy. The AFDB Swiss-Prot dataset is a subset of the AlphaFold Protein Structure Database that specifically contains structure predictions for all protein sequences corresponding to UniProt Swiss-Prot entries. Currently, the dataset includes

Table 17: Hyperparameter setting of the ProteinBERT pre-training.

| MLM | |
|---|---|
| Batch Size | 48 |
| Learning Rate | 1e-4 |
| Warmup Ratio | 0.05 |
| Mask Ratio | 0.15 |
| Hidden Dimension | 512 |
| Layer Depth | 12 |
| MVCL | |
| Batch Size | 96 |
| Learning Rate | 1e-4 |
| Warmup Ratio | 0.05 |
| Subsequence Length | 50 |
| Max Node | 50 |
| Temperature | 0.0099 |
| Hidden Dimension | 512 |
| Layer Depth | 12 |
| PFP | |
| Batch Size | 48 |
| Learning Rate | 1e-4 |
| Warmup Ratio | 0.05 |
| Temperature | 0.0099 |
| Hidden Dimension | 512 |
| Layer Depth | 12 |

a total of 542,378 entries. ESM-2 and SaProt are pretrained on UniRef sequences clustered at the 50% sequence identity level. ESM-C is also pretrained on UniRef but with clustering at the 70% sequence identity level, and further incorporates protein sequences from the MGnify and Joint Genome Institute (JGI) databases, resulting in 83M, 372M, and 2B clusters for UniRef, MGnify, and JGI, respectively.

**Settings**. We adopt task and architecture-specific training and model hyperparameters across different pretraining tasks and model types. Detailed configurations are summarized in Tab. 13-17.

For the Multi-View Contrastive Learning (MVCL) task, we set the maximum length of subsequences to 50. To extract substructures in 3D space, we randomly sample a central amino acid and collect all residues within a 15 Å Euclidean radius, with the subspace length capped at 50 residues. Temperature values used in the contrastive objective are adapted for each model architecture to ensure stable training dynamics.

For the Protein Family Prediction (PFP) task, we address the sparsity of family labels by adopting a negative sampling strategy. During training, family labels are embedded into the same representation space as proteins, and the model is optimized to align protein embeddings with their corresponding family embeddings. Simultaneously, embeddings of proteins from other families within the same batch are pushed apart, following a contrastive learning framework. A temperature coefficient is introduced to scale the similarity scores and enhance training stability.

