# OpenReview forum: "Protap: A Benchmark for Protein Modeling on Realistic Downstream Applications"
_ICLR.cc/2026/Conference — Submitted to ICLR 2026_

### Official Review · Reviewer_Jhsw · 2025-10-17

**Soundness:** 2
**Presentation:** 2
**Contribution:** 2
**Rating:** 2
**Confidence:** 5

**Summary:**

Here, the authors assemble a benchmark suite to evaluate pretrained protein models. They conduct an extensive grid evaluation of different pretraining strategies across different architectures, pretraining models on a subset of proteins from UniRef that overlap with protein structures from the AlphaFold Protein Database.

**Strengths:**

Overall, the high-level sketch of this work is interesting and impactful: effectively, the authors are seeking to do a comprehensive evaluation of different pretraining strategies and architectures, across different kinds of downstream tasks. What differentiates this from previous similar benchmarks would be:

1) The inclusion of three (very) differentiated pretraining tasks. Most benchmarks only include tasks based upon language modeling (e.g. masked language or autoregressive models), but previous evidence has noted that these tasks may scale sub-optimally for some kinds of downstream applications. The inclusion of tasks that use different kinds of signal than recovering missing amino acids in sequences is interesting especially in light of calls from these prior works to explore diversified pretraining tasks.

2) Similarly, a much more diversified range of model architectures than previous works, including models that use structural priors compared to models that operate on sequence-only.

3) The comparison of pre-trained protein models to domain-specific models. This sets a higher standard for benchmarking and is relevant given the observation that pre-trained protein models fail to outperform randomly initialized baselines or regression on one-hot encoded sequences.

**Weaknesses:**

However, there are numerous issues in the actual execution of this work, that lead to an overall poor evaluation from me.

1) There is likely substantial leakage between one of the pre-training strategies and one of the downstream tasks. In the protein family prediction pretraining task, models are pre-trained to predict family labels from Uniprot. Critically, we can expect families to be either subsets of, or in some cases, effectively entirely overlap gene ontology function classes in the protein function annotation prediction task. This is caused by the fact that the Gene Ontology Consortium uses phylogenetic inference and sequence homology to transfer functional annotations, which overlap with the same strategies that Uniprot use to assign protein families.

2) Standardizing pretraining data across pretraining tasks occludes the way that these tasks have different generality and scalability. Pretraining tasks that rely upon structure or annotation are necessarily limited to smaller pretraining datasets than what is possible for sequence-only pretraining tasks. Furthermore, sequence-only models have been previously documented to rely upon implicit learning of evolutionary homology for their effectiveness, which is only possible if the models have been exposed to many homologs during pretraining - so training them on filtered-down datasets where only sequences where structure is available may hurt their effectiveness a lot more than if they were trained on larger datasets relative to models that explicitly incorporate priors. In other words, it is difficult to fully disentangle pretraining tasks from data because pretraining tasks are designed to scale and learn from data differently - and critically, this work is intended to make a general statement about pretraining strategies in general (i.e. which strategies are more effective), as opposed to e.g. pretraining strategies in low-data regimes.

3) Many critical details are omitted from the manuscript, and it is impossible to evaluate whether the benchmarking has been done properly without these details. There are many cases of this, but the ones that stood out to me were:
- There is no information on how long models were trained for, or what stopping criterion was used. This is not reported in Supplement D, which only documents hyperparameter settings. Currently, all of the pre-trained models perform fairly poorly compared to randomly initialized equivalents on most tasks (with the exception of the protein functional annotation task for the protein family prediction pretrained models, but I noted that this can be explained by leakage), and it is not clear if the authors have trained these models to sufficient convergence to draw the conclusion that pretraining is often ineffective.
- The description of the pretraining tasks is incomplete and confusing, and details are omitted from even the supplementary. For example, for the multi-view contrastive learning pretraining task, I cannot tell if this is on subsequences, substructures, or subsequences plus substructure. The section starts by saying the task "preserves the representational similarity between the correlated substructures of proteins", which suggests learning on structure, but then subsequently says "we generate two views... from its amino acid sequence as positive samples", which suggest learning on sequence. But the views are denoted differently, and in Figure 1a, Gseq and Gspace are represented differently (with arrows connecting Gspace) suggesting Gseq is sequence and Gspace is structure. There is no formal definition of the pretraining task even in the supplementary, so I cannot figure out what this task actually is.
- There is limited explanation of the pretraining data curation, and this is especially critical for the protein family prediction task. The authors explain that they take structures from APDB and cross-reference these against Uniprot to collect sequence and family labels. But what level of evidence in family labels do they consider? What databases do they collect identifiers from? What does the distribution of family labels look like (e.g. if they're considering all databases/identifiers this will likely be sparse and very high-dimensional - again, exactly what is the pretraining task, is it a cross-entropy classification problem over all of these)? Again, there is no detail in supplementary about this.

4) The comparisons of Protap to previous benchmark suites are overstated:
- In particular, Table 1 aims to frame Protap as the benchmark that covers the most applications, heavily suggesting that their benchmark is comprehensive compared to others. But this table is curated to be specific to features present in Protap, but does not include any features present in other benchmark suites but not present in Protap - as the authors themselves note, other benchmarks cover aspects like generative design, protein engineering, protein-protein interactions, or simply aim for depth of coverage over a specific topic over breadth. In other words - I think it is inaccurate to frame Protap as comprehensive compared to previous efforts, because it is clear these efforts are complementary.
- Second, at least three of the five datasets are from prior machine learning efforts where substantial benchmarking work has already been done for the individual dataset in the original paper. This is in contrast to many previous benchmarks, which curate datasets from large-scale bioinformatics databases, existing experimental data, or even collect experimental data by themselves. I raise this because in contrast to these works, much less translational effort has been put into the curation of this benchmark e.g. instead of needing to format and clean messy data for machine learning purposes or make decisions on training/test splits, this benchmarking effort primarily recycles this existing work done by other authors. I'm uncomfortable with the assertion that these applications have not been thoroughly studied in "existing benchmarks", because even as this work is comprehensive, these applications have been benchmarked by the original authors that curated the dataset.

**Questions:**

Overall, the issues I raise in weaknesses are substantial enough that I think it would be challenging to comprehensively address them all within a short rebuttal period - especially because of the lack of complete information in the original manuscript, it is possible that clarifications of these questions will raise further issues. But to begin with:

1. Provide full details on pretraining datasets, pretraining losses, and exactly how models were trained.
2. Provide a stratification/split of the function classification task such that there is no overlap and homology with pretraining data.
3. Train the sequence and function pretrained models on larger datasets more reflective of the scale of data these pretraining tasks would typically interact with, as opposed to structure-scale datasets.
4. Reduce the claim that Protap is comprehensive compared to previous benchmark suites, and more accurately portray it complementary, and be more explicit about how this benchmark recycles existing datasets that have already been thoroughly evaluated with baselines (although not the specific strategies/baselines in this work).

---

> ### Author Response · Authors · 2025-11-22
> **Response to Reviewer Jhsw**
>
> Thank you for the detailed review and thoughtful feedback. I will provide some clarification regarding the weaknesses and questions you mentioned. Due to the character limit, the remaining issues will be answered in the following response.
>
> >W1: Potential leakage from Pfam family information into GO-based function prediction".
>
> A: The Pfam information we use during pretraining includes only: which representative Pfam domains (i.e., family IDs) are contained in each sequence. Specifically, the Pfam domain definitions we use originate from InterPro. To give one concrete example, PF00168 (C2 domain) is explicitly defined based on the 3D conformation of the domain and its calcium-binding pocket. Pfam is based on the structural domain conservation at the sequence level, rather than any GO functional rules or functional propagation mechanisms.
>
> In fact, for protein function prediction tasks, prior SOTA works such as DPFunc have already used InterPro structural region information to enhance function prediction. Their motivation is to leverage domain information to guide the model in learning the functional relevance of amino acids within their structural context, thereby highlighting structure regions that are highly associated with function.
>
> To elaborate further: Pfam family annotations and GO functional annotations belong to different ontology systems and are not semantically equivalent. Pfam is a structural-domain–based conservation system and is part of InterPro domain resources. Its definitions are fundamentally based on: evolutionary conservation of sequence domains, structural fold patterns, MSA and HMM-based domain boundary detection. **Pfam refers to a structurally conserved domain region, not a functional category as defined in GO.**
>
> In contrast, GO annotations describe molecular function, biological process, and cellular component, typically involving: experimental literature evidence, phylogenetic inference, homology-based transfer.
>
> Although some statistical correlation may exist because domains often participate in particular functions, the two systems are semantically independent, and there is no mapping from GO to Pfam. Therefore, Pfam labels do not directly leak GO functional categories. Regarding the reviewer’s concern that using domain-based family information might introduce functional leakage, we believe this should not be framed as a clear-cut weakness, but rather as an open question worthy of further discussion in the community.
>
> >W2 & Q3: "Standardizing pretraining data may distort conclusions because different pretraining tasks inherently scale differently, making it hard to separate task effectiveness from data availability”.
>
> A2: To ensure that our conclusions are not biased by limited data, and to explicitly examine whether MLM-style protein pretraining exhibits scaling laws, we directly incorporate large scale sequence PLMs whose pretraining corpora reflect realistic usage:
>
> * ESM-2 (UR50 70M)
> * SaProt (UR50 40M)
> * ESM-C (UR70 3B)
>
> **These models were pretrained on tens of millions of sequences** orders of magnitude larger than any structure derived dataset. Thus, our evaluation already covers the high data regime where sequence pretraining is known to be strongest.
>
> **ProtBert on Swiss-Prot 540k is used intentionally as a controlled study to isolate pretraining-task effects**
>
> To fairly compare different pretraining tasks and architectures, we include a controlled-scale experiment:
>
> | Analysis Type        | Purpose                                                          | Model and Data scale                                                      |
> |----------------------|------------------------------------------------------------------|------------------------------------------------------------|
> | Controlled Study     | Use a unified data size to fairly compare pretraining tasks and architectures | ProteinBert (Swiss-Prot 540k)                                         |
> | At-scale Model Study | Compare SOTA sequence-pretrained models                 | ESM-2 (UR50, 70M), SaProt (UR50, 40M), ESM-C (UR70, JDI, 3B) |
>
>
> Using the same 540k-scale comparison provides controlled conditions to answer an orthogonal question: “do different pretraining objectives behave differently?” Using a structure-scale dataset for ProtBert allows us to directly compare the performance of different pretraining strategies without confounding the dataset size.
>
> Through this controlled experimental setup, our findings reveal an important insight:
>
> Although MLM pretraining has excellent scalability, even with tens of millions of training sequences, the learned signal still fails to encode the cross entity geometric information required for PLI/PROTAC/PCS tasks. This reflects a structural limitation of current pretraining objectives, rather than a limitation of data volume.

---

> ### Author Response · Authors · 2025-11-22
> **Response to Reviewer Jhsw**
>
> Here, we address your Weakness 3 and Question 1.
>
> > W3 &Q1: "Many critical details are omitted from the manuscript, such as details about pretrain method and train process".
>
> A3: **For the first sub-question** "regarding whether the pretrained models may not have converged":
>
> I would like to first emphasize that randomly initialized equivalent models must be trained from scratch on data, rather than being randomly initialized and directly evaluated. Second, during training we use a uniform number of steps across models, ensuring comparable optimization budgets. We have released all pretraining and downstream task code, and further implementation details can be found in the repository. In the revised version, we will additionally include loss curves in the appendix to make convergence behavior explicit.
>
> **For the second sub-question** "detail of the training procedure of MVCL":
>
> Multi-View Contrastive Learning uses both sequence and structure to construct local subviews of the same protein under the assumption that “subspaces and subsequences of the same protein should share similar high-level representations”. This assumption was proposed in GearNet.
>
> We generate two types of subviews:
>
> 1. Subsequence view (𝐺seq)
>    Randomly crop a continuous fragment of length subseq\_length using get\_subsequence. This is a position based subsequence.
> 2. Subspace view (𝐺space)
>    Pick a central residue, collect neighbors within Euclidean radius d, truncate to max\_nodes or zero-pad (get\_subspace). This is a 3D neighborhood-based subspace view.
>
> For each protein, we randomly sample two views (subseq+subseq,subspace+subspace, or subseq+subspace), then encode them and get representation.
>
> We train using the symmetric InfoNCE loss:
>
> $$\\mathcal{L}\_{\\text{MVCL}}=\\frac{1}{2B}\\sum\_{i=1}^{2B}-\\log\\frac{\\exp(\\mathrm{sim}(z\_i, z\_{i^+}) / \\tau)}{\\sum\_{j \\ne i}\\exp(\\mathrm{sim}(z\_i, z\_j) / \\tau)}$$
>
> where $\tau$ is the temperature parameter.
>
> **For the third sub-question** "the construction of the pretraining dataset and detail of PFP":
>
> As I answered in W1, The Pfam information we use during pretraining includes only: which representative Pfam domains (i.e., family IDs) are contained in each sequence. Specifically, the Pfam domain definitions we use originate from InterPro.
>
> Protein Family Prediction is implemented as a contrastive multi-label prediction task. Specifically:
>
> Protein representations and family embeddings: For each protein, we obtain a graph-level representation$h_i$  from the encoder. The model also maintains a family embedding matrix $E \in \mathbb{R}^{K \times d}$, where each row $e_k$ corresponds to the embedding of a Pfam family.
>
> Positive labels and padding strategy: Each protein is associated with a multi-label family set $P_i$. In the implementation, the first *pos_num* entries of *batch_family* are true family IDs, and the remaining entries are padded with $-100$.
>
> Negative sampling: For each protein, the padded positions are replaced with randomly sampled negative families drawn from families not in $P_i$, forming the negative set $N_i$.
>
> Contrastive classification for each positive family: For every positive family $p \in P_i$, we construct a “1 positive + multiple negatives’’ softmax contrastive task:
> - Positive logit: $\mathrm{sim}(h_i, e_p)$
> - Negative logits: $\{\mathrm{sim}(h_i, e_n)\}_{n \in N_i}$
>
> Final loss for PFP:
>
> \\(\\mathcal{L}\_{\\text{PFP}}=\\frac{1}{\\sum\_i|P\_i|}\\sum\_i\\sum\_{p\\in P\_i}-\\log\\frac{\\exp(\\mathrm{sim}(h\_i,e\_p)/\\tau)}{\\exp(\\mathrm{sim}(h\_i,e\_p)/\\tau)+\\sum\_{n\\in N\_i}\\exp(\\mathrm{sim}(h\_i,e\_n)/\\tau)}\\)
>
> where:
> \\(h\_i\\) is the protein graph representation,
> \\(e\_p\\) is the embedding of a positive family,
> \\(N\_i\\) is the set of sampled negative families for protein \\(i\\),
> \\(\\tau\\) is a temperature hyperparameter.
>
> As described above, we enumerate all positive families for each protein and construct an independent contrastive subtask for each positive label. This design helps partially alleviate the strong class imbalance in Pfam family annotations.
>
> We will include this more complete and formal definition of the MVCL and PFP objective in the appendix of the revised version.

---

> ### Author Response · Authors · 2025-11-22
> **Response to Reviewer Jhsw**
>
> In this response, I will continue addressing the remaining issues.
>
> >W4 & Q4: "The comparisons of Protap to previous benchmark suites are overstated".
>
> A4: We agree with your point regarding the term comprehensive, the term complementary is indeed more appropriate. Our selection of evaluation tasks was not designed to be exhaustive, but rather to focus on realistic applications and to benchmark tasks that are more aligned with industrial relevance. We will reduce the claim of “comprehensive” in future revisions.
>
> At the same time, we would like to clarify that Table 1 is accurate. As shown in the table, protein function prediction and mutation effect prediction have been evaluated in other benchmarks. However, to our knowledge, no existing domain-specialized benchmark has evaluated the domain-specific tasks on the datasets we use. Therefore, our statement that “previous benchmarks have not sufficiently evaluated these domain-specific tasks” is accurate. We welcome additional references you may suggest.
>
> Regarding your suggestions about data sources and reuse, we provide the following clarification:
>
> As stated in the main text, we did not collect downstream data ourselves from scattered sources; instead, we used datasets released by prior work, and applied preprocessing to standardize them into the unified Protap data format for training, evaluation, and future extensibility.
>
> For each downstream task, the dataset origins are clearly stated and cited in Table 2(b) of the main text. In addition, Section D of the appendix further documents the original data sources and licenses for each task. For datasets originating from other benchmarks, for example, mutation effect prediction and protein function prediction, we explicitly list the benchmark names (ProteinGym and ProteinWorkshop) and provide the links to the original hosts. Except for the ProteinGym mutation dataset, all other downstream tasks (including protein function prediction) suffer from missing coordinates, missing structures, or inconsistent protein identifiers. Therefore, we performed substantial preprocessing, detailed in Section D (Pre-process). After preprocessing, the dataset sizes are slightly reduced compared to the originals; details are shown in Table 2(b).
>
> >Q2: "Risk of information leakage between pretraining and downstream tasks".
>
> A2: During pretraining, our objectives are entirely unsupervised, and no downstream task labels are ever used; therefore, there is no possibility of label leakage.
>
> For downstream evaluations, such as protein function prediction and protein cleavage site prediction,we apply explicit dataset splits and ensure that sequence similarity between training and test sets is kept below the appropriate thresholds, preventing any overlap that could cause information leakage.

---

### Official Review · Reviewer_wErr · 2025-10-21

**Soundness:** 3
**Presentation:** 3
**Contribution:** 2
**Rating:** 4
**Confidence:** 3

**Summary:**

This paper presents Protap, a new benchmark for protein modeling that systematically evaluates a wide range of backbone architectures, pretraining strategies, and domain-specific models on a suite of five realistic downstream protein applications. Protap encompasses both widely studied general tasks and two specialized, industrially relevant tasks (enzyme-catalyzed protein cleavage site prediction and targeted protein degradation via PROTACs) that are largely absent from prior benchmarks. The paper offers a holistic comparison across architectures (including sequence-based, structure-based, and hybrid models), pretraining regimes (MLM, multi-view CL, protein family prediction), and domain-specific models, reporting extensive experiments and analysis on standardized datasets.

**Strengths:**

1. Protap covers a broad swath of both general and specialized protein tasks, filling gaps unaddressed by leading prior benchmarks.
2. The experimental design is comprehensive, testing a wide array of architectures and several pretraining objectives. The study further incorporates and details strong domain-specific models for specialized tasks.
3. The paper presents performance tables comparing scores of all baseline and domain-specific models using standard metrics, and provides setup for all experiments, including details on hyperparameters and data preprocessing. Results are reported clearly with mean and standard deviation over multiple seeds, and ablations are provided.
4. All code, data, and pipeline details are made available, with careful documentation.

**Weaknesses:**

1. The experimental setup is extensive, but direct benchmarking versus leading results from highly cited works on individual tasks (especially broader benchmarks like ProtTrans, MSA Transformer) is missing. For example, for mutation effect and function prediction, existing strong baselines from recent works are absent as explicit baselines in the main results tables. This makes it harder to quantitatively situate the new results relative to the current state of the art in every aspect.
2. While the experimental results are detailed, the investigation into why certain architectures excel in specific contexts is somewhat superficial, at times reduced to high-level hypotheses about inductive bias. The potential confounding effects of hyperparameter choices, data overlap, or domain-specific tuning are not exhaustively ruled out.
3. The modeling of the non-protein components in multimodal interaction tasks is oversimplified. In tasks involving interactions between proteins and other molecules (such as Protein-Ligand Interaction (PLI) and PROTACs), the final predictive performance depends on the ability to represent all participating components, not just the protein.

**Questions:**

1. For MVCL and PFP pretraining, can the authors specify the exact loss functions, sampling (especially for negatives in MVCL), and any rebalancing strategies used? Please clarify if temperature scaling is applied and how the family-imbalance in PFP is handled.
2. The observation in Table 4b is very interesting: on the PROTACs degradation prediction task, the general-purpose EGNN model outperforms two domain-specific models (DeepPROTACS, ET-PROTACS). This is a counter-intuitive result. Could you provide some potential hypotheses to explain this phenomenon?
3. In tasks involving small molecules, such as PLI and PROTACs, the general-purpose protein models were uniformly configured with a GVP encoder to handle the small molecules. What was the rationale for this design choice? Could this have systematically lowered the performance ceiling of the general-purpose models on these tasks, thereby affecting the fairness of the comparison against domain-specific models, which may have more optimized molecular representation modules? Did you consider or try using a standard, pre-trained molecular encoder as a stronger baseline?
4. The main text of the paper primarily presents and discusses the results from frozen-encoder fine-tuning, while the results for full-parameter fine-tuning, a common paradigm, are placed in the appendix. Could you explain the reasoning behind this organizational choice? Given that the full-parameter fine-tuning results in the appendix (Table 6) show a more complex phenomenon (with mixed performance, sometimes even showing a decrease), would this alter or supplement the main conclusions you draw in the main text regarding the value of the pre-trained model?

---

> ### Author Response · Authors · 2025-11-22
> **Response to Reviewer wErr**
>
> We sincerely thank the reviewer for the careful reading and the constructive comments. Below, we address a subset of your points in detail. Due to the character limit, the remaining issues will be answered in the following response.
>
> >W: "Protap lacks direct comparison to several established SOTA baselines (e.g., ProtTrans, MSA Transformer), making it harder to contextualize its results".
>
> A1: Although the goal of our work is to investigate how different pretraining strategies, data scales, and domain-specific inductive biases behave across diverse downstream tasks, rather than to identify a new SOTA mode, we would like to clarify that, in practice, for all downstream tasks except mutation effect prediction, our chosen baselines are in fact the most influential or SOTA methods available on the corresponding evaluation datasets.
>
> For example:
>
> * PFA: we use DeepFunc, a leading method on this task.
> * PCS: we include Unizyme, the SOTA model in PCS task.
> * PROTACs: we incorporate ET-PROTAC, also the SOTA for this task.
>
> For protein–ligand interaction (PLI), the current SOTA models are those that perform large-scale PLI-specific pretraining (e.g., DrugCLIP). However, such methods are pretrained on other large PLI corpora and therefore cannot be compared fairly under a unified setup. Hence, we use KDBNet, which is the SOTA model on the kinase–ligand interaction dataset used in our benchmark.
>
> In addition, the external pretrained models we include ESM-2, ESM-Cambrian, and SaProt which represent some of the most recent and influential large-scale pretrained protein models, all of which have demonstrated strong performance across many evaluations, and whose representations are widely adopted in downstream protein modeling tasks.
>
> >W2: "Explanations for why certain architectures perform better remain shallow and do not rule out factors like hyperparameters, data overlap, or domain-specific tuning".
>
> A2: First, the importance of structural information is indeed our central explanation. For example, in RQ3, we observe that structural models (e.g., EGNN) outperform sequence-only models (e.g., ProtBERT) on interaction-centric tasks such as PROTACs. The most direct explanation is that structural inductive biases are critical for modeling 3D intermolecular interactions, whereas sequence models inherently lack such geometry-aware priors. Moreover, when comparing pretrained models across modalities, we find that pretraining is beneficial only when the downstream task aligns with the pretraining objective (e.g., single sequence evolutionary modeling). In contrast, when the downstream task requires multi-entity relational signals and structural docking information that are absent from the pretraining corpus, the pretrained representations fail to transfer effectively again highlighting the central role of structural information.
>
> To rule out potential confounding factors, we standardize the downstream training setup across all models (same optimizer, learning-rate schedule, and five random seeds), thereby controlling for hyperparameter effects. We also use datasets with strict sequence-identity splits (e.g., PFA \< 50%) to prevent overlap between train and test sets. In addition, the reproduced performance of domain-specific models closely matches the original papers, providing further evidence that our experimental conclusions are robust and reliable.

---

> ### Author Response · Authors · 2025-11-22
> **Response to Reviewer wErr**
>
> In this response, I will explain Question 1 and Question 2 that you mentioned.
>
> >Q1: "Details of MVCL and PFP pretraining, such as loss functions and sampling strategies".
>
> A1: Thank you for highlighting the need for clearer pretraining descriptions. Although our released code contains full details, we will provide explicit formulas and sampling procedures in the next version. The formal definitions are as follows.
>
> **MVCL (Multi-View Contrastive Learning)**
>
> MVCL uses both sequence and structure to construct local subviews of the same protein under the assumption that
>
> “subspaces and subsequences of the same protein should share similar high-level representations”. This assumption was proposed in GearNet.
>
> We generate two types of subviews:
>
> 1. Subsequence view (𝐺seq)
>    Randomly crop a continuous fragment of length subseq\_length using get\_subsequence. This is a position based subsequence.
> 2. Subspace view (𝐺space)
>    Pick a central residue, collect neighbors within Euclidean radius d, truncate to max\_nodes or zero-pad (get\_subspace). This is a 3D neighborhood-based subspace view.
>
> For each protein, we **randomly sample two views** (subseq+subseq,subspace+subspace, or subseq+subspace), then encode them and get representation.
>
> We train using the symmetric InfoNCE loss:
>
> $$\\mathcal{L}\_{\\text{MVCL}}=\\frac{1}{2B}\\sum\_{i=1}^{2B}-\\log\\frac{\\exp(\\mathrm{sim}(z\_i, z\_{i^+}) / \\tau)}{\\sum\_{j \\ne i}\\exp(\\mathrm{sim}(z\_i, z\_j) / \\tau)}$$
>
> where $\tau$ is the temperature parameter.
>
> **PFP (Protein Family Prediction)**
>
> PFP pretraining is implemented as a contrastive multi-label prediction task. Specifically:
>
> Protein representations and family embeddings: For each protein, we obtain a graph-level representation$h_i$  from the encoder. The model also maintains a family embedding matrix $E \in \mathbb{R}^{K \times d}$, where each row $e_k$ corresponds to the embedding of a Pfam family.
>
> Positive labels and padding strategy: Each protein is associated with a multi-label family set $P_i$. In the implementation, the first *pos_num* entries of *batch_family* are true family IDs, and the remaining entries are padded with $-100$.
>
> Negative sampling: For each protein, the padded positions are replaced with randomly sampled negative families drawn from families not in $P_i$, forming the negative set $N_i$.
>
> Contrastive classification for each positive family: For every positive family $p \in P_i$, we construct a “1 positive + multiple negatives’’ softmax contrastive task:
> - Positive logit: $\mathrm{sim}(h_i, e_p)$
> - Negative logits: $\{\mathrm{sim}(h_i, e_n)\}_{n \in N_i}$
>
> Final loss for PFP:
>
> \\(\\mathcal{L}\_{\\text{PFP}}=\\frac{1}{\\sum\_i|P\_i|}\\sum\_i\\sum\_{p\\in P\_i}-\\log\\frac{\\exp(\\mathrm{sim}(h\_i,e\_p)/\\tau)}{\\exp(\\mathrm{sim}(h\_i,e\_p)/\\tau)+\\sum\_{n\\in N\_i}\\exp(\\mathrm{sim}(h\_i,e\_n)/\\tau)}\\)
>
> where:
> \\(h\_i\\) is the protein graph representation,
> \\(e\_p\\) is the embedding of a positive family,
> \\(N\_i\\) is the set of sampled negative families for protein \\(i\\),
> \\(\\tau\\) is a temperature hyperparameter.
>
> As described above, we enumerate all positive families for each protein and construct an independent contrastive subtask for each positive label. This design helps partially alleviate the strong class imbalance in Pfam family annotations.
>
> We will include this more complete and formal definition of the MVCL and PFP objective in the appendix of the revised version.
>
> >Q2: "Why do EGNN models outperform DeepPROTACs and ET-PROTAC in PROTACs task?".
>
> A2: We carefully analyzed the architectures of DeepPROTACs and ET-PROTAC. The key difference lies in how protein 3D structure is treated:
>
> 1. DeepPROTACs use a 2D GCN, losing 3D geometric information.
> 2. ET-PROTAC uses a CNN encoder, also discarding residue-level spatial geometry.
> 3. EGNN, in contrast, preserves full 3D-equivariant relational structure and performs geometric message passing.
>
> PROTAC modeling is a tri-partite spatial problem (E3 ligase, POI, linker), and spatial arrangement is crucial. EGNN explicitly models geometric constraints, long-range interactions and equivariant transformations.
>
> These are precisely the inductive biases that DeepPROTACs and ET-PROTAC lack, explaining why EGNN achieves significantly better performance.

---

> ### Author Response · Authors · 2025-11-22
> **Response to Reviewer wErr**
>
> In this response, I will continue addressing the remaining issues.
>
> >Q3 & W3: "Why do all general-purpose protein models use a GVP encoder for small-molecule representation instead of more complex molecular encoders?".
>
> A3: The reason we use GVP as the molecular encoder and train it from scratch is precisely to enable a fair comparison. For the tasks we introduce that involve molecular modeling, the domain-specific models also use GVP as their molecular encoder, so using GVP for all general-purpose models makes the comparison fair and consistent. If we switch to a pretrained molecular encoder, such as a more complex encoder like Uni-Mol, it would change the original model architecture and design; if we only change the molecular encoder corresponding to each pretrained protein encoder, it becomes difficult to determine whether the performance improvement or degradation comes from the protein encoder or the molecular encoder when compared to domain-specific models. In addition, GVP has relatively few parameters, and our PROTAC training dataset is small, introducing a more complex molecular encoder may lead to underfitting on the molecular side. Moreover, by keeping the molecular encoder unified, although this may slightly reduce the overall performance of all models, it does not affect our comparison of model performance on this task.
>
> >Q4: “The reason for placing frozen-encoder results in the main text and putting full fine-tuning results in Table 6”
>
> A4: Thanks for your careful observation. First, I would like to address your question regarding the placement and organization of the tables. Indeed, this work primarily presents the fine-tuning results under the frozen-encoder setting in the main text, while placing the full-parameter fine-tuning experiments in the appendix. This organizational choice is based on the following considerations:
>
> 1. Benchmark fairness and comparability: The benchmark covers five realistic downstream applications, some of which have limited data (e.g., enzyme cleavage and PROTAC). In these scenarios, full-parameter fine-tuning exhibits large differences in gradient sensitivity across models, making optimization instability more likely. In contrast, freezing the encoder and training only the task head produces a more reproducible and comparable evaluation across different model scales and architectures. Therefore, we chose to highlight and discuss these frozen-encoder results in the main paper.
> 2.  Alignment with the research question on pretraining transferability: The core question of this paper is: Do pretrained representations provide consistent transfer benefits across downstream tasks? The frozen-encoder setting preserves pretrained knowledge while minimizing parameter disturbance, making it better suited to answer our key question: Are the structural/semantic priors learned during pretraining reusable in downstream tasks? For this reason, prioritizing frozen-encoder results in the main text better reflects the central research objective.
>
> Below, I will explain the reasons behind the fluctuations and performance degradation observed during full-parameter fine-tuning.
>
> The performance drops observed in some full-parameter fine-tuning results in Appendix Table 6 are expected and intentionally reported. We hypothesize that these phenomena stem from:
>
> * semantic mismatch between downstream data and the pretraining distribution;
> * full-parameter updates in large PLMs disrupting evolution-based priors shared across protein families;
> * sequence-only PLMs failing to adapt to structure-sensitive tasks (such as PLI and PCS task) without structural supervision.
>
> Therefore, these results do not weaken the conclusions in the main paper. Instead, they further support our claims regarding the limitations of pretraining transfer.
>
> After re-examining Table 6, we confirm that although full-parameter fine-tuning yields more complex behaviors, it does not alter the conclusions in the main text. On the contrary, it complements them by providing additional evidence. These observations provide important references for protein modeling tasks. To avoid obscuring the main evaluation narrative, we chose to place full-parameter fine-tuning results in the appendix.

---

### Official Review · Reviewer_R4Qj · 2025-10-27

**Soundness:** 2
**Presentation:** 3
**Contribution:** 2
**Rating:** 2
**Confidence:** 4

**Summary:**

The paper presents Protap, a benchmark that evaluates protein modeling approaches across five realistic applications: cleavage site prediction, PROTAC-mediated degradation, protein–ligand interaction, function annotation, and mutation effect prediction. Protap provides a unified framework to assess trade-offs between general foundation models and task-specific designs. By comparing pretrained protein language models, structure-based and hybrid architectures, and domain-specific models, the study reports that training from scratch can sometimes outperform frozen large encoders, structural information often boosts downstream performance, and domain-specific models dominate in specialized applications.

**Strengths:**

The benchmark scope is extensive along three axes tasks, architectures, and pre-training strategies, and it includes two biologically meaningful specialized applications that are under represented in prior benchmarks. The protocol describes pre training data sources, downstream splits, and multi seed reporting, which improves reproducibility.

**Weaknesses:**

- Lack of cross-benchmark synthesis: While Protap positions itself against previous benchmarks, it does not compare its findings with prior benchmarks that consistently reported strong benefits of pretraining on downstream tasks. In contrast, Protap observes that training from scratch often outperforms pretrained models. The paper does not explain what differences in setup (e.g., task selection, optimization regime, data scale) might account for this contradiction.
- Domain-specific model advantage not fully contextualized: While domain models often outperform pretrained PLMs in specialized applications, the paper does not clearly provide deeper insights into the underlying reasons.
- Possible under-optimization of pretrained encoders or fine-tuning process: The weaker performance of pretrained encoders may be due to insufficient optimization of pre-training or limited exploration of fine-tuning strategies. The paper does not present ablations to rule out these possibilities.
- Ambiguity in molecular encoders: Section 3.1 states that molecular components are represented by randomly initialized GVP encoders, but it is unclear whether these encoders are trained downstream. This could significantly impact performance.

**Questions:**

- Previous benchmarks reported benefits of pretraining on downstream tasks, whereas Protap often finds the opposite. What differences in setup explain this contradiction?
- In cases where domain-specific models outperform general pretrained models, can the authors provide deeper insights into the underlying reasons for this advantage?
- Did the authors conduct ablations to ensure that pre-trained encoders or fine-tuning process are not under-optimized?
- For the molecular encoders, are the randomly initialized GVP encoders trained during downstream experiments, and if so, under what settings?

---

> ### Author Response · Authors · 2025-11-22
> **Response to Reviewer R4Qj**
>
> Thank you for your detailed feedback. Due to the page limit, I will address the first two weaknesses and questions in this response, and I will address the remaining concerns in the following response.
>
> >W1 & Q1: "Prior benchmarks generally report benefits from pretraining, whereas Protap often observes the opposite. What explains this discrepancy?".
>
> A1: First, I would like to clarify our interpretation of the results and the conclusion we draw from the tables: pretraining is beneficial when the downstream task resembles the pretraining objective (i.e., single sequence evolutionary modeling), but when a downstream task requires multi-entity relational information and structural docking signals that are absent from the pretraining corpus, the pretrained knowledge fails to transfer effectively.
>
> Therefore, our results do not imply that “pretraining is ineffective”; rather, they indicate that current mainstream pretraining objectives whether sequence level MLM or existing structure contrastive objectives do not encode the cross entity geometric reasoning required by the domain-specialized, multi-component tasks in Protap, which leads to limited transferability in these settings.
>
> Concretely, Earlier benchmarks such as PEER and ProteinWorkshop predominantly evaluate single-protein tasks (e.g., GO function prediction, enzyme family classification, subcellular localization). Protap also includes two such single-input tasks—Protein Function Annotation (PFA) and Mutation Effect Prediction (MTP).
>
> As shown clearly in Tables 3 and 6, pretrained models outperform scratch models in most single-protein settings. This is exactly the finding summarized in A2, Observation 3:
>
> “Large-scale models such as SaProt, ESM-2 and ESM-C demonstrate clear advantages on tasks requiring only a single protein sequence as input, such as PFA and MTP.”
>
> Thus, our results are not contradictory to prior work. Instead, we confirm previous conclusions in single-protein modeling.
>
> In contrast, Protap’s specialized downstream tasks include multi-entity tasks (PLI, PROTAC, PCS), which depend heavily on structural relationships across multiple molecular components. For example：
>
> * protein–ligand spatial complementarity
> * ternary PROTAC assembly
> * enzyme–substrate interface alignment
> * multi-chain relative orientation
>
> However，Our benchmark includes structure based pretraining (MVCL, PFP), yet these objectives primarily learn:
>
> * intra-protein local geometry
> * domain-level evolutionary conservation
> * single-chain structural context
>
> they still do not expose the model to inter-molecular geometry. Thus, even structure-pretrained models exhibit limited transfer to these task types. In contrast, scratch-trained geometric encoders (e.g., EGNN/GVP) directly optimize on the multi-entity spatial objective and therefore learn the cross-body geometric inductive biases that current pretraining objectives do not provide.
>
> This explains why Protap observes weaker pretraining transfer on multi-entity tasks, while fully matching prior benchmarks on single sequence tasks. We will make this distinction explicit in the revised version.
>
> >W2 & Q2: "Deeper analysis explaining why domain-specific models outperform general PLMs in  specific tasks".
>
> A2: We summarize this in RQ2 and A2 (Observations 3 and 4).
>
> Below, we provide additional direct examples of the domain-specific designs used by SOTA methods:
>
> 1. Unizyme (PCS task) incorporates energy frustration priors into the attention module in enzyme modeling, adding biologically grounded constraints. This is a bio-aware architectural prior unavailable to general PLMs.
> 2. DeepFunc (PFA task) uses InterProScan derived domain annotations to highlight structure-function regions, guiding the model to functionally meaningful amino-acid subregions.
> 3. KDBNet (PLI task) uses ESM embeddings only as part of node initialization for a graph neural network, rather than a final representation. The model further integrates multiple node features and performs structured message passing.
>
> These examples suggest two overarching principles:
>
> 1. Task-specific biological priors matter: domain knowledge—enzyme energetics, structural domains, binding geometric constraints—substantially improves specialization.
> 2. PLM embeddings are more effective as initialization than as final frozen features: combining PLM representations with structured architectures accelerates convergence and improves accuracy compared with simple concatenation into a prediction head.

---

> ### Author Response · Authors · 2025-11-22
> **Response to Reviewer R4Qj**
>
> In this response, I will continue addressing the remaining issues.
>
> > W3 & Q3: "Possible under-optimization of pretrained encoders or fine-tuning process".
>
> A3: We provide the following evidence eliminating this concern. We will also include the loss curves in the supplementary materials of the subsequent revised version.
>
> 1. All models both pretrained and scratch show fully converged loss curves during pretraining and fine-tuning.
> 2. ESM-2, SaProt and ESM Cambrian models are not trained by us, we directly used the authors’ released weights. Their downstream behavior is consistent with our own pretrained models, indicating optimization was not a limiting factor.
> 3. For domain-specific models, our reproduced results closely match the original papers, which further confirms the training pipeline has converged properly.
>
> Given that both our pretrained models and external pretrained models show consistent patterns, the reduced transfer performance is unlikely to arise from under optimization.
>
> >W4 & Q4: "Clarification about molecular encoder training.
>
> A4: Thank you again for highlighting this. All molecular GVP encoders are randomly initialized and fully trained during downstream optimization. The implementation is available in our open source codebase for more detail.

---

### Official Review · Reviewer_i3yn · 2025-10-29

**Soundness:** 3
**Presentation:** 4
**Contribution:** 2
**Rating:** 6
**Confidence:** 4

**Summary:**

Protap introduces a benchmarking suite for evaluating general‑purpose and domain‑specific protein models on five realistic downstream applications: two specialized tasks (enzyme‑catalyzed protein cleavage site prediction and PROTAC‑based targeted protein degradation) and three general tasks (protein‑ligand interaction prediction, protein function annotation, and mutation effect prediction). The benchmark systematically compares a wide spectrum of architectures (sequence‑based Transformers, geometric graph neural networks, sequence-structure hybrid models), pre‑training strategies (masked‑language modeling, multi‑view contrastive learning and protein family prediction), and domain‑specific models across these tasks.

**Strengths:**

- Prior benchmarks such as PEER and TAPE focused on general sequence or structural tasks; none considered enzyme cleavage or PROTAC‑mediated degradation.

- The paper clearly outlines four concrete observations drawn from its benchmark evaluation: (1) pretraining improves downstream performance but gains vary significantly across tasks; (2) task-specific models can outperform general-purpose PLMs when domain adaptation is well-aligned with task distribution; (3) there is low cross-task correlation, indicating that high performance in one application (e.g., protein fitness prediction) does not generalize to others (e.g., PROTAC efficacy); and (4) current models struggle particularly with enzyme cleavage site prediction, suggesting this task remains an open challenge. These observations are well-motivated, empirically supported, and help ground the benchmark’s value beyond raw leaderboard comparisons.

- The benchmark evaluates a broad range of architectures, from sequence language models (ESM‑2, ProtBERT) to geometric GNNs (EGNN, SE(3) Transformer, GVP) and hybrid models (SaProt, D‑Transformer). It also includes domain‑specific models tailored to each task (e.g., UniZyme for cleavage, DeepProtacs and ET‑PROTACs for PROTACs, KDBNet and MONN for ligand interactions). This breadth helps researchers understand how architectural choices and inductive biases affect performance across tasks.

- The results demonstrate that models incorporating 3D structural information (EGNN, SE(3) Transformer, GVP, D‑Transformer) generally outperform pure sequence models. Even when pre‑trained, geometric models achieve stronger performance on protein–ligand interaction and PROTAC tasks. The authors argue that structures provide essential inductive biases absent in sequence‑only architectures.

- Table 4 (page 9) compares general models to eight domain‑specific models. General architectures such as EGNN and D‑Transformer match or exceed specialized models for PROTACs, while specialized models like KDBNet outperform general models on protein–ligand interaction. For protein function annotation, DeepFRI and DPFunc (models that incorporate structural and functional information) significantly outperform general models. The benchmark, therefore, highlights the contexts where specialized inductive biases are necessary.

- The authors release the code and data, including hyper‑parameter settings and pre‑training details, enabling the community to reproduce and extend the experiments.

**Weaknesses:**

- While the benchmark adds enzyme cleavage and PROTAC tasks, it still omits other important applications such as protein folding, stability prediction, de novo protein design, antigen–antibody binding and protein–protein interactions beyond binary classification. Benchmarks like ProteinBench and ProteinGym include tasks for protein design and conformational dynamics, whereas Protap focuses mainly on classification/regression problems. Extending the benchmark to cover additional tasks would significantly increase its impact.

- The specialized tasks use relatively small datasets: the enzyme cleavage task has 375 training proteins and 92 test proteins, and the PROTAC task has 343 training pairs and 202 test pairs. Small sample sizes may lead to high variance and limit the generalizability of conclusions. In addition, the annotation quality of enzyme cleavage sites and PROTAC ternary complexes varies across sources; the benchmark does not discuss potential noise or biases in these datasets.

- Performance is measured by metrics such as AUC, AUROC and MSE. For tasks like cleavage site prediction and PROTAC degradation, false negatives can have very different practical implications than false positives (e.g., missing a cleavage site may be more problematic than incorrectly predicting one). Moreover, the reported metrics are cost-agnostic and insensitive to imbalance in ways that can mask real-world risk. In cleavage-site prediction (and similarly in PROTAC degradation screens), a false negative (missing a true site/degrader) is often far more damaging than a false positive, yet AUROC/AUC weight FP and FN equally. So two models with the same AUROC can have radically different miss rates at the FP budgets labs can afford.

- The benchmark compares models of varying sizes (10M to 650M parameters) but does not investigate scaling laws. Even though the authors observe that smaller models can outperform larger ones (e.g., EGNN vs. ESM‑2), they do not explore whether performance improves with moderate scale (ESM-2 family) or how parameter counts relate to data efficiency. ProteinBench emphasises scaling laws for protein representation models; Protap could benefit from a similar analysis.

**Questions:**

- Your RQ1 suggests that encoders trained from scratch often outperform or match pre-trained PLMs (under frozen or linear-head settings). However, your conclusion may not be directly comparable: the examples you highlight often compare a pre-trained sequence model with a supervised structure-based model, rather than comparing models within the same modality. Within-modality trends may actually agree, e.g., ProteinWorkshop (ICLR’24) shows pre-training does benefit structure-based models. Could your “scratch > pre-train” finding simply reflect cross-modality gaps or domain-task mismatch, rather than a failure of pre-training itself? Would you expect the conclusions to change if you compared pre-trained vs scratch within each modality (e.g., a pre-trained structure GNN vs supervised one), or if you fully fine-tuned the sequence models under the same protocol?

- You evaluate MLM-style and related pre-training but appear not to include next-token (autoregressive) pre-training; even though AR protein LMs (e.g., ProGen-style families) are used beyond generation (zero-shot mutation effect via Δlog-likelihood/perplexity, and embeddings from hidden states with simple pooling). Could you clarify why AR objectives were excluded?


**Suggestions for improvement**
- Expand task coverage: Incorporate additional downstream tasks such as protein–protein interaction prediction, antigen/antibody binding, protein folding stability and generative design. Including these tasks would align Protap more closely with benchmarks like ProteinGym and ProteinBench.

- Increase dataset size and diversity: Augment the enzyme cleavage and PROTAC datasets by integrating additional experimental sources and negative examples to reduce sampling bias. Where feasible, leverage high‑throughput assays or simulated data to enlarge the training sets.

- Investigate scaling laws and parameter efficiency: Systematically vary model size within each architecture (e.g., small, medium, large EGNN/SE(3)/Transformer) to understand how performance scales with parameters and data. This could reveal whether moderate scaling yields better trade‑offs than extremely large language models.

- Explore multi‑task and cross‑task learning: Evaluate whether jointly training a model on multiple Protap tasks improves generalization, following the findings from PEER. Additionally, investigate whether knowledge learned from specialized tasks (e.g., cleavage prediction) transfers to general tasks (e.g., function annotation).


If the authors address the weaknesses outlined above, or provide a right justification for the points raised in my questions, or incorporate any of the suggested future directions or clarify points I may have misunderstood, I would be willing to raise my score to an 8.

---

> ### Author Response · Authors · 2025-11-22
> **Response to Reviewer i3yn**
>
> Thank you for your kind recognition of our work and for offering valuable feedback. Due to the page limit, I will address your weaknesses in this response, and I will address your question in the next response.
>
> > W1: “The benchmark can include more downstream tasks.”
>
> A1: On the one hand, we agree with your perspective that a comprehensive benchmark could incorporate an even broader set of downstream tasks to yield insights across more application scenarios. We also plan to maintain Protap as a long-term evolving project and to include additional protein understanding tasks in future releases, such as interface and binding-site prediction, immunogenicity prediction, and other tasks closely aligned with real-world applications.
>
> On the other hand, I would like to emphasize that Protap focuses specifically on protein understanding tasks. ProteinBench focuses on design and structure generation, and ProteinGym focuses on mutation effect prediction. These benchmarks address different problem domains. As you suggested, the additional tasks we plan to incorporate, such as antigen–antibody binding and structure prediction, will continue to follow the theme of protein understanding.
>
> > W2: “The specialized tasks rely on small and potentially noisy datasets and potential noise or biases may influence results”.
>
> A2: We would like to provide two clarifications regarding dataset scale and quality:
>
> 1. Cleave-site prediction: Although the dataset contains 343 training proteins, each protein typically includes multiple proteolytic sites. The detailed cleave site statistics are as follows: the M10003 family contains 878 training samples and 157 test samples, and the C14005 family contains 641 training samples and 157 test samples. Furthermore, our evaluation uses two enzyme families, which helps mitigate biases associated with any single dataset source.
> 2. PROTAC prediction: As detailed in Appendix D.2, the dataset contains 4,122 training \+ test samples, rather than the 545 samples mentioned in the review.
>
> Additionally, all datasets undergo strict quality control filtering steps, which are fully described in Appendix D.
>
> >W3: “Cost-insensitive metrics on PROTACs and PCS tasks”.
>
> A3: We fully agree that cost-insensitive metrics such as AUROC may fail to reflect real world FN/FP asymmetry in biochemical screening. We clarify the following:
>
> 1. Cleave-site prediction uses AUPR as the primary metric, since this residue level classification task is highly imbalanced. AUPR is more sensitive to FN/recall, aligning with the practical need to avoid missing true cleavage sites.
> 2. PROTAC prediction is formulated as a balanced binary classification problem following prior literature. As shown in D.2, the positive/negative ratio is not skewed; therefore, we adopt Accuracy and AUC, consistent with standard practice in previous PROTACs prediction studies.
>
> >W4: "Performance comparison of pretrained models with different parameter scales".
>
> A4: We appreciate this suggestion and also noted this limitation explicitly in our Conclusion as valuable future work. Given the importance of this insight, we conducted additional experiments comparing ESM2 models of 35M, 150M, and 650M parameters across PROTAC, PLI, and PCS tasks. The results are shown below:
> | Train Setting   | Size | C14.005 | C14.005 | PROTACDB | PROTACDB | Davis | Davis |
> |-----------------|------|---------|---------|----------|----------|--------|--------|
> |                 |      | AUC (%)↑ | AUPR (%)↑ | Acc (%)↑ | AUC (%)↑ | MSE ↓ | Pear (%)↑ |
> | fine-tuning     | 35M  | 80.38±0.77 | 87.53±1.09 | 0.4908±0.0022 | 52.81±0.74 | 96.40±0.01 | 40.09±0.20 |
> | fine-tuning     | 150M | 81.40±1.37 | 87.45±0.96 | 0.4935±0.0129 | 52.46±1.38 | 96.44±0.02 | 42.33±0.93 |
> | fine-tuning     | 650M | 78.46±3.03 | 85.74±3.47 | 0.4910±4.823  | 53.25±1.26 | 97.23±0.06 | 41.22±0.25 |
> | Freeze encoder  | 35M  | 82.85±2.22 | 90.01±1.12 | 0.4569±0.0091 | 57.32±1.56 | 95.21±1.26 | 47.92±1.07 |
> | Freeze encoder  | 150M | 82.25±0.35 | 89.21±0.39 | 0.4589±0.0029 | 57.07±0.02 | 90.57±3.52 | 50.04±6.24 |
> | Freeze encoder  | 650M | 81.25±1.44 | 88.17±1.10 | 0.4160±0.0110 | 59.59±0.35 | 93.10±1.71 | 55.09±0.22 |
>
> These results reinforce our original conclusion: frozen (encoder-only) ESM2 models consistently outperform full fine-tuning, and freezing yields more stable improvements across model sizes.
>
> Moreover, under the frozen-encoder setting, the 650M ESM2 model achieves the best results on PLI and PCS, indicating that scaling laws do exist for sequence models in protein representation learning.

---

> ### Author Response · Authors · 2025-11-22
> **Response to Reviewer i3yn**
>
> In this response, I address the questions raised in your review.
>
> >Q1: "RQ1 conclusion may be misleading because it is based on cross-modality comparisons, which are inherently unfair".
>
> A1: Our experiments include not only cross-modality comparisons, but also within-modality comparisons, as well as within-modality analyses across different pretraining data scales:
>
> **1\. Within-sequence-modality comparison** (from-scratch vs. pretrained language models)
>
> Because our from-scratch models include a pure sequence encoder (ProtBert), we are able to analyze sequence models in isolation from structure-based models. Within the sequence modality, we examine two dimensions:
>
> • From-scratch vs. pretrained: ProtBert (scratch) vs. ProtBert (pretrained)
>
> • Pretraining at different data scales: ProtBert (pretrained on 540k) vs. ESM-2 (pretrained on tens of millions)
>
> From this comparison, we observe that pretraining consistently improves performance on single-protein tasks (PFA, MTP), but brings limited or no gains for PROTAC, cleavage, and PLI. Thus, even within the same modality, MLM-based sequence pretraining does not reliably transfer to geometry-dependent tasks.
>
> **2\. Within-structure-modality comparison**
>
> We also perform controlled evaluations within the structure modality, again isolating structural encoders from sequence models.
>
> * Structure encoders trained from scratch: EGNN, SE(3)-Transformer, GVP
> * Three types of structure-based pretraining model
>
> We find structure pretraining yields only marginal or no improvements. Thus, the pattern “scratch ≥ pretrain” also holds within the structure modality.
>
> **3\. Cross-modality comparisons** quantify the benefit of adding structure Under matched data conditions and encoder design:
>
> * EGNN (trained from scratch) \> ProtBert (trained from scratch), demonstrating that 3D structural information plays a critical role.
> * EGNN (trained from scratch) \> ESM-2 / SaProt, indicating that even large-scale sequence pretraining cannot compensate for the absence of structural information.
>
> This indicates that the performance differences arise from the inherent dependence of tasks such as PROTAC and PLI on structural information for multi-entity interaction modeling, highlighting the **importance of structure** rather than any failure of pretraining itself.
>
> >Q2: "Why autoregressive models are not included among the pretrained models".
>
> A2: We initially considered including autoregressive (AR) models for comparison, but ultimately decided not to for two main reasons.
>
> 1. First, many of our downstream tasks require global representations or representations for all residues (e.g., the PCS task). Due to the unidirectional and sequential nature of AR decoding, the representation of the i-th residue can only incorporate information from positions ≤ i−1. This is a significant limitation for tasks such as PCS, which require modeling of bidirectional context around cleavage sites. Prior work has also shown that BERT-style bi-directional models outperform AR models on protein understanding tasks which requiring global representations. For example, ProtTrans explicitly notes:
>    “**Bi-directionality is crucial for pLMs. In NLP, uni-directional and bi-directional models perform on par, but in proteins the bi-directional context appears essential to model aspects of the language of life.**”
> 2. Second, our benchmark focuses on protein understanding tasks (PROTAC,PCS, PLI, PFA, MTP), many of which require multi-chain reasoning and structural context. AR protein language models, however, are primarily designed for sequence generation, not for multi-entity relational understanding, and thus are not well aligned with the goal of Protap.
>
> For these two reasons, we ultimately chose not to include autoregressive models.

---

### Meta-Review · Area_Chair_8Kvf · 2026-01-05

**Summary:**

Based on the reviews and the authors' rebuttal, I recommend **Rejecting** this paper.


The paper introduces "Protap," a benchmark designed to evaluate protein modeling across five downstream applications, including three general tasks (function, mutation, ligand interaction) and two specialized, industrially relevant tasks (enzyme cleavage, PROTAC degradation). The authors benchmark various architectures (sequence, structure, hybrid) and pretraining strategies, concluding that training from scratch often outperforms large-scale pretraining on specialized tasks and that structural priors are crucial for multi-entity interactions.

**Strengths:**
The reviewers universally appreciated the motivation to include specialized, realistic tasks like PROTACs and enzyme cleavage, which are missing from existing benchmarks like TAPE or PEER. The inclusion of diverse architectures and the release of code/data are also positives.

**Reasons for Rejection:**
However, significant concerns remain regarding the execution and validity of the benchmark results, which were not fully resolved during the rebuttal:

1.  **Validity of Conclusions ("Scratch > Pretrain"):** A major point of contention (Reviewers R4Qj, wErr) is the finding that supervised models trained from scratch outperform pretrained models.

2.  **Potential Data Leakage:** Reviewer Jhsw raised a critical concern regarding leakage between the "Protein Family Prediction" (PFP) pretraining task and the "Protein Function Annotation" (PFA) downstream task.

3.  **Dataset and Baseline Limitations:** The specialized datasets (e.g., PROTACs) are relatively small, raising questions about the statistical robustness of the comparisons.

4.  **Novelty:** As noted by Reviewer Jhsw, the benchmark largely aggregates existing datasets rather than generating new data or curation methods.

While the direction of benchmarking complex biochemical tasks is promising, the methodological flaws regarding optimization, baselines, and potential leakage need to be addressed before publication. Besides, the anonymous URL for code and data is not valid while I process this submission.

**Reviewer Concerns:**

**Reasons for Rejection:**
However, significant concerns remain regarding the execution and validity of the benchmark results, which were not fully resolved during the rebuttal:

1.  **Validity of Conclusions ("Scratch > Pretrain"):** A major point of contention (Reviewers R4Qj, wErr) is the finding that supervised models trained from scratch outperform pretrained models. While the authors argue this is due to task misalignment (single-sequence pretraining vs. multi-entity downstream tasks), reviewers remain concerned that this result stems from suboptimal fine-tuning or hyperparameter selection for the pretrained models. The fact that full-parameter fine-tuning results were relegated to the appendix—and often showed degradation—suggests potential optimization issues rather than a fundamental limitation of pretraining. A benchmark must demonstrate SOTA optimization to make such strong claims.

2.  **Potential Data Leakage:** Reviewer Jhsw raised a critical concern regarding leakage between the "Protein Family Prediction" (PFP) pretraining task and the "Protein Function Annotation" (PFA) downstream task. While the authors argue that Pfam (structural) and GO (functional) are distinct ontologies, they are biologically highly correlated. Pretraining on families and testing on function without rigorous homology filtering or split stratification undermines the validity of the results for this specific track.

3.  **Dataset and Baseline Limitations:** The specialized datasets (e.g., PROTACs) are relatively small, raising questions about the statistical robustness of the comparisons. Furthermore, key baselines (e.g., ProtTrans, MSA Transformer) were missing, and the molecular encoders were simplified (randomly initialized GVPs), which may artificially lower the ceiling for general-purpose models compared to domain-specific ones.

4.  **Novelty:** As noted by Reviewer Jhsw, the benchmark largely aggregates existing datasets rather than generating new data or curation methods. Without rigorous experimental setups that align with community standards (e.g., resolving the pretraining effectiveness debate definitively), the aggregation alone is insufficient for acceptance.

**Reviewer Scores:**

The four reviewers assigned scores of **6, 4, 2, and 2**.

The overall feedback is predominantly negative with an average score of 3.5.
*   **Reviewer i3yn** gave a **6** (marginally above threshold), appreciating the benchmark's breadth.
*   **Reviewer wErr** gave a **4** (marginally below threshold).
*   **Reviewers R4Qj and Jhsw** both gave **2** (strong reject), raising critical concerns about the validity of the conclusions (specifically the "scratch > pretrain" claim), potential data leakage, and suboptimal experimental setups.

Given the two strong rejections and low average, the consensus leans heavily toward rejection.

---

### Decision · Program_Chairs · 2026-01-26

Reject